# Moment-based Uniform Deviation Bounds for $k$-means and Friends

**Matus Telgarsky**      **Sanjoy Dasgupta**
Computer Science and Engineering, UC San Diego
{mtelgars,dasgupta}@cs.ucsd.edu

## Abstract

Suppose $k$ centers are fit to $m$ points by heuristically minimizing the $k$-means cost; what is the corresponding fit over the source distribution? This question is resolved here for distributions with $p \geq 4$ bounded moments; in particular, the difference between the sample cost and distribution cost decays with $m$ and $p$ as $m^{\min\{-1/4, -1/2+2/p\}}$. The essential technical contribution is a mechanism to uniformly control deviations in the face of unbounded parameter sets, cost functions, and source distributions. To further demonstrate this mechanism, a soft clustering variant of $k$-means cost is also considered, namely the log likelihood of a Gaussian mixture, subject to the constraint that all covariance matrices have bounded spectrum. Lastly, a rate with refined constants is provided for $k$-means instances possessing some cluster structure.

## 1   Introduction

Suppose a set of $k$ centers $\{p_i\}_{i=1}^k$ is selected by approximate minimization of $k$-means cost; how does the fit over the sample compare with the fit over the distribution? Concretely: given $m$ points sampled from a source distribution $\rho$, what can be said about the quantities

$$\left| \frac{1}{m} \sum_{j=1}^m \min_i \|x_j - p_i\|_2^2 - \int \min_i \|x - p_i\|_2^2 d\rho(x) \right| \qquad (k\text{-means}), \qquad (1.1)$$

$$\left| \frac{1}{m} \sum_{j=1}^m \ln \left( \sum_{i=1}^k \alpha_i p_{\theta_i}(x_j) \right) - \int \ln \left( \sum_{i=1}^k \alpha_i p_{\theta_i}(x) \right) d\rho(x) \right| \qquad (\text{soft } k\text{-means}), \qquad (1.2)$$

where each $p_{\theta_i}$ denotes the density of a Gaussian with a covariance matrix whose eigenvalues lie in some closed positive interval.

The literature offers a wealth of information related to this question. For $k$-means, there is firstly a consistency result: under some identifiability conditions, the global minimizer over the sample will converge to the global minimizer over the distribution as the sample size $m$ increases [1]. Furthermore, if the distribution is bounded, standard tools can provide deviation inequalities [2, 3, 4]. For the second problem, which is maximum likelihood of a Gaussian mixture (thus amenable to EM [5]), classical results regarding the consistency of maximum likelihood again provide that, under some identifiability conditions, the optimal solutions over the sample converge to the optimum over the distribution [6].

The task here is thus: to provide finite sample guarantees for these problems, but eschewing boundedness, subgaussianity, and similar assumptions in favor of moment assumptions.

## 1.1 Contribution

The results here are of the following form: given $m$ examples from a distribution with a few bounded moments, and any set of parameters beating some fixed cost $c$, the corresponding deviations in cost (as in eq. (1.1) and eq. (1.2)) approach $\mathcal{O}(m^{-1/2})$ with the availability of higher moments.

- In the case of $k$-means (cf. Corollary 3.1), $p \geq 4$ moments suffice, and the rate is $\mathcal{O}(m^{\min\{-1/4,-1/2+2/p\}})$. For Gaussian mixtures (cf. Theorem 5.1), $p \geq 8$ moments suffice, and the rate is $\mathcal{O}(m^{-1/2+3/p})$.

- The parameter $c$ allows these guarantees to hold for heuristics. For instance, suppose $k$ centers are output by Lloyd's method. While Lloyd's method carries no optimality guarantees, the results here hold for the output of Lloyd's method simply by setting $c$ to be the variance of the data, equivalently the $k$-means cost with a single center placed at the mean.

- The $k$-means and Gaussian mixture costs are only well-defined when the source distribution has $p \geq 2$ moments. The condition of $p \geq 4$ moments, meaning the variance has a variance, allows consideration of many heavy-tailed distributions, which are ruled out by boundedness and subgaussianity assumptions.

The main technical byproduct of the proof is a mechanism to deal with the unboundedness of the cost function; this technique will be detailed in Section 3, but the difficulty and its resolution can be easily sketched here.

For a single set of centers $P$, the deviations in eq. (1.1) may be controlled with an application of Chebyshev's inequality. But this does not immediately grant deviation bounds on another set of centers $P'$, even if $P$ and $P'$ are very close: for instance, the difference between the two costs will grow as successively farther and farther away points are considered.

The resolution is to simply note that there is so little probability mass in those far reaches that the cost there is irrelevant. Consider a single center $p$ (and assume $x \mapsto \|x - p\|_2^2$ is integrable); the dominated convergence theorem grants

$$\int_{B_i} \|x - p\|_2^2 d\rho(x) \quad \to \quad \int \|x - p\|_2^2 d\rho(x), \qquad \text{where } B_i := \{x \in \mathbb{R}^d : \|x - p\|_2 \leq i\}.$$

In other words, a ball $B_i$ may be chosen so that $\int_{B_i^c} \|x - p\|_2^2 d\rho(x) \leq 1/1024$. Now consider some $p'$ with $\|p - p'\|_2 \leq i$. Then

$$\int_{B_i^c} \|x - p'\|_2^2 d\rho(x) \leq \int_{B_i^c} (\|x - p\|_2 + \|p - p'\|_2)^2 d\rho(x) \leq 4 \int_{B_i^c} \|x - p\|_2^2 d\rho(x) \leq \frac{1}{256}.$$

In this way, a single center may control the outer deviations of whole swaths of other centers. Indeed, those choices outperforming the reference score $c$ will provide a suitable swath. Of course, it would be nice to get a sense of the size of $B_i$; this however is provided by the moment assumptions.

The general strategy is thus to split consideration into outer deviations, and local deviations. The local deviations may be controlled by standard techniques. To control outer deviations, a single pair of dominating costs — a lower bound and an upper bound — is controlled.

This technique can be found in the proof of the consistency of $k$-means due to Pollard [1]. The present work shows it can also provide finite sample guarantees, and moreover be applied outside hard clustering.

The content here is organized as follows. The remainder of the introduction surveys related work, and subsequently Section 2 establishes some basic notation. The core deviation technique, termed *outer bracketing* (to connect it to the bracketing technique from empirical process theory), is presented along with the deviations of $k$-means in Section 3. The technique is then applied in Section 5 to a soft clustering variant, namely log likelihood of Gaussian mixtures having bounded spectra. As a reprieve between these two heavier bracketing sections, Section 4 provides a simple refinement for $k$-means which can adapt to cluster structure.

All proofs are deferred to the appendices, however the construction and application of outer brackets is sketched in the text.

## 1.2 Related Work

As referenced earlier, Pollard's work deserves special mention, both since it can be seen as the origin of the outer bracketing technique, and since it handled $k$-means under similarly slight assumptions (just two moments, rather than the four here) [1, 7]. The present work hopes to be a spiritual successor, providing finite sample guarantees, and adapting technique to a soft clustering problem.

In the machine learning community, statistical guarantees for clustering have been extensively studied under the topic of *clustering stability* [4, 8, 9, 10]. One formulation of stability is: if parameters are learned over two samples, how close are they? The technical component of these works frequently involves finite sample guarantees, which in the works listed here make a boundedness assumption, or something similar (for instance, the work of Shamir and Tishby [9] requires the cost function to satisfy a bounded differences condition). Amongst these finite sample guarantees, the finite sample guarantees due to Rakhlin and Caponnetto [4] are similar to the development here *after* the invocation of the outer bracket: namely, a covering argument controls deviations over a bounded set. The results of Shamir and Tishby [10] do not make a boundedness assumption, but the main results are not finite sample guarantees; in particular, they rely on asymptotic results due to Pollard [7].

There are many standard tools which may be applied to the problems here, particularly if a boundedness assumption is made [11, 12]; for instance, Lugosi and Zeger [2] use tools from VC theory to handle $k$-means in the bounded case. Another interesting work, by Ben-david [3], develops specialized tools to measure the complexity of certain clustering problems; when applied to the problems of the type considered here, a boundedness assumption is made.

A few of the above works provide some negative results and related commentary on the topic of uniform deviations for distributions with unbounded support [10, Theorem 3 and subsequent discussion] [3, Page 5 above Definition 2]. The primary "loophole" here is to constrain consideration to those solutions beating some reference score $c$. It is reasonable to guess that such a condition entails that a few centers must lie near the bulk of the distribution's mass; making this guess rigorous is the first step here both for $k$-means and for Gaussian mixtures, and moreover the same consequence was used by Pollard for the consistency of $k$-means [1]. In Pollard's work, only optimal choices were considered, but the same argument relaxes to arbitrary $c$, which can thus encapsulate heuristic schemes, and not just nearly optimal ones. (The secondary loophole is to make moment assumptions; these sufficiently constrain the structure of the distribution to provide rates.)

In recent years, the empirical process theory community has produced a large body of work on the topic of maximum likelihood (see for instance the excellent overviews and recent work of Wellner [13], van der Vaart and Wellner [14], Gao and Wellner [15]). As stated previously, the choice of the term "bracket" is to connect to empirical process theory. Loosely stated, a bracket is simply a pair of functions which sandwich some set of functions; the *bracketing entropy* is then (the logarithm of) the number of brackets needed to control a particular set of functions. In the present work, brackets are paired with sets which identify the far away regions they are meant to control; furthermore, while there is potential for the use of many outer brackets, the approach here is able to make use of just a single outer bracket. The name bracket is suitable, as opposed to cover, since the bracketing elements need not be members of the function class being dominated. (By contrast, Pollard's use in the proof of the consistency of $k$-means was more akin to covering, in that remote fluctuations were compared to that of a a single center placed at the origin [1].)

## 2 Notation

The ambient space will always be the Euclidean space $\mathbb{R}^d$, though a few results will be stated for a general domain $\mathcal{X}$. The source probability measure will be $\rho$, and when a finite sample of size $m$ is available, $\hat{\rho}$ is the corresponding empirical measure. Occasionally, the variable $\nu$ will refer to an arbitrary probability measure (where $\rho$ and $\hat{\rho}$ will serve as relevant instantiations). Both integral and expectation notation will be used; for example, $\mathbb{E}(f(X)) = \mathbb{E}_\rho(f(X) = \int f(x)d\rho(x)$; for integrals, $\int_B f(x)d\rho(x) = \int f(x)\mathbb{1}[x \in B]d\rho(x)$, where $\mathbb{1}$ is the indicator function. The moments of $\rho$ are defined as follows.

**Definition 2.1.** *Probability measure $\rho$ has* order-$p$ moment bound $M$ with respect to norm $\|\cdot\|$ *when* $\mathbb{E}_\rho\|X - \mathbb{E}_\rho(X)\|^l \leq M$ *for $1 \leq l \leq p$.*

For example, the typical setting of $k$-means uses norm $\|\cdot\|_2$, and at least two moments are needed for the cost over $\rho$ to be finite; the condition here of needing 4 moments can be seen as naturally arising via Chebyshev's inequality. Of course, the availability of higher moments is beneficial, dropping the rates here from $m^{-1/4}$ down to $m^{-1/2}$. Note that the basic controls derived from moments, which are primarily elaborations of Chebyshev's inequality, can be found in Appendix A.

The $k$-means analysis will generalize slightly beyond the single-center cost $x \mapsto \|x - p\|_2^2$ via *Bregman divergences* [16, 17].

**Definition 2.2.** *Given a convex differentiable function* $f : \mathcal{X} \to \mathbb{R}$, *the corresponding* Bregman divergence *is* $\mathsf{B}_f(x, y) := f(x) - f(y) - \langle \nabla f(y), x - y \rangle$.

Not all Bregman divergences are handled; rather, the following regularity conditions will be placed on the convex function.

**Definition 2.3.** *A convex differentiable function* $f$ *is* strongly convex *with modulus* $r_1$ *and has* Lipschitz gradients *with constant* $r_2$, *both respect to some norm* $\|\cdot\|$, *when* $f$ *(respectively) satisfies*

$$f(\alpha x + (1-\alpha)y) \leq \alpha f(x) + (1-\alpha)f(y) - \frac{r_1 \alpha(1-\alpha)}{2}\|x-y\|^2,$$

$$\|\nabla f(x) - \nabla f(y)\|_* \leq r_2 \|x - y\|,$$

*where* $x, y \in \mathcal{X}$, $\alpha \in [0, 1]$, *and* $\|\cdot\|_*$ *is the dual of* $\|\cdot\|$. *(The Lipschitz gradient condition is sometimes called* strong smoothness.*)*

These conditions are a fancy way of saying the corresponding Bregman divergence is sandwiched between two quadratics (cf. Lemma B.1).

**Definition 2.4.** *Given a convex differentiable function* $f : \mathbb{R}^d \to R$ *which is strongly convex and has Lipschitz gradients with respective constants* $r_1, r_2$ *with respect to norm* $\|\cdot\|$, *the* hard $k$-means cost *of a single point* $x$ *according to a set of centers* $P$ *is*

$$\phi_f(x; P) := \min_{p \in P} \mathsf{B}_f(x, p).$$

*The corresponding* $k$-means cost *of a set of points (or distribution) is thus computed as* $\mathbb{E}_\nu(\phi_f(X; P))$, *and let* $\mathcal{H}_f(\nu; c, k)$ *denote all sets of at most* $k$ *centers beating cost* $c$, *meaning*

$$\mathcal{H}_f(\nu; c, k) := \{P : |P| \leq k, \mathbb{E}_\nu(\phi_f(X; P)) \leq c\}.$$

For example, choosing norm $\|\cdot\|_2$ and convex function $f(x) = \|x\|_2^2$ (which has $r_1 = r_2 = 2$), the corresponding Bregman divergence is $\mathsf{B}_f(x, y) = \|x - y\|_2^2$, and $\mathbb{E}_{\hat\rho}(\phi_f(X; P))$ denotes the vanilla $k$-means cost of some finite point set encoded in the empirical measure $\hat\rho$.

The hard clustering guarantees will work with $\mathcal{H}_f(\nu; c, k)$, where $\nu$ can be either the source distribution $\rho$, or its empirical counterpart $\hat\rho$. As discussed previously, it is reasonable to set $c$ to simply the sample variance of the data, or a related estimate of the true variance (cf. Appendix A).

Lastly, the class of Gaussian mixture penalties is as follows.

**Definition 2.5.** *Given Gaussian parameters* $\theta := (\mu, \Sigma)$, *let* $p_\theta$ *denote Gaussian density*

$$p_\theta(x) = \frac{1}{\sqrt{(2\pi)^d |\Sigma_i|}} \exp\left(-\frac{1}{2}(x - \mu_i)^T \Sigma_i^{-1}(x - \mu_i)\right).$$

*Given Gaussian mixture parameters* $(\alpha, \Theta) = (\{\alpha_i\}_{i=1}^k, \{\theta_i\}_{i=1}^k)$ *with* $\alpha \geq 0$ *and* $\sum_i \alpha_i = 1$ *(written* $\alpha \in \Delta$*), the Gaussian mixture cost at a point* $x$ *is*

$$\phi_{\mathrm{g}}(x; (\alpha, \Theta)) := \phi_{\mathrm{g}}(x; \{(\alpha_i, \theta_i) = (\alpha_i, \mu_i, \Sigma_i)\}_{i=1}^k) := \ln\left(\sum_{i=1}^k \alpha_i p_{\theta_i}(x)\right),$$

*Lastly, given a measure* $\nu$, *bound* $k$ *on the number of mixture parameters, and spectrum bounds* $0 < \sigma_1 \leq \sigma_2$, *let* $\mathcal{S}_{\mathrm{mog}}(\nu; c, k, \sigma_1, \sigma_2)$ *denote those mixture parameters beating cost* $c$, *meaning*

$$\mathcal{S}_{\mathrm{mog}}(\nu; c, k, \sigma_1, \sigma_2) := \{(\alpha, \Theta) : \sigma_1 I \preceq \Sigma_i \preceq \sigma_2 I, |\alpha| \leq k, \alpha \in \Delta, \mathbb{E}_\nu(\phi_{\mathrm{g}}(X; (\alpha, \Theta))) \leq c\}.$$

While a condition of the form $\Sigma \succeq \sigma_1 I$ is typically enforced in practice (say, with a Bayesian prior, or by ignoring updates which shrink the covariance beyond this point), the condition $\Sigma \preceq \sigma_2 I$ is potentially violated. These conditions will be discussed further in Section 5.

# 3 Controlling $k$-means with an Outer Bracket

First consider the special case of $k$-means cost.

**Corollary 3.1.** *Set $f(x) := \|x\|_2^2$, whereby $\phi_f$ is the $k$-means cost. Let real $c \geq 0$ and probability measure $\rho$ be given with order-$p$ moment bound $M$ with respect to $\| \cdot \|_2$, where $p \geq 4$ is a positive multiple of 4. Define the quantities*

$$c_1 := (2M)^{1/p} + \sqrt{2c}, \quad M_1 := M^{1/(p-2)} + M^{2/p}, \quad N_1 := 2 + 576d(c_1 + c_1^2 + M_1 + M_1^2).$$

*Then with probability at least $1 - 3\delta$ over the draw of a sample of size $m \geq \max\{(p/(2^{p/4+2}e))^2, 9\ln(1/\delta)\}$, every set of centers $P \in \mathcal{H}_f(\hat{\rho}; c, k) \cup \mathcal{H}_f(\rho; c, k)$ satisfies*

$$\left| \int \phi_f(x; P)d\rho(x) - \int \phi_f(x; P)d\hat{\rho}(x) \right|$$

$$\leq m^{-1/2 + \min\{1/4, 2/p\}} \left( 4 + (72c_1^2 + 32M_1^2)\sqrt{\frac{1}{2}\ln\left(\frac{(mN_1)^{dk}}{\delta}\right)} + \sqrt{\frac{2^{p/4}ep}{8m^{1/2}}}\left(\frac{2}{\delta}\right)^{4/p} \right).$$

One artifact of the moment approach (cf. Appendix A), heretofore ignored, is the term $(2/\delta)^{4/p}$. While this may seem inferior to $\ln(2/\delta)$, note that the choice $p = 4\ln(2/\delta)/\ln(\ln(2/\delta))$ suffices to make the two equal.

Next consider a general bound for Bregman divergences. This bound has a few more parameters than Corollary 3.1. In particular, the term $\epsilon$, which is instantiated to $m^{-1/2+1/p}$ in the proof of Corollary 3.1, catches the mass of points discarded due to the outer bracket, as well as the resolution of the (inner) cover. The parameter $p'$, which controls the tradeoff between $m$ and $1/\delta$, is set to $p/4$ in the proof of Corollary 3.1.

**Theorem 3.2.** *Fix a reference norm $\| \cdot \|$ throughout the following. Let probability measure $\rho$ be given with order-$p$ moment bound $M$ where $p \geq 4$, a convex function $f$ with corresponding constants $r_1$ and $r_2$, reals $c$ and $\epsilon > 0$, and integer $1 \leq p' \leq p/2 - 1$ be given. Define the quantities*

$$R_B := \max\left\{(2M)^{1/p} + \sqrt{4c/r_1}, \max_{i\in[p']}(M/\epsilon)^{1/(p-2i)}\right\},$$

$$R_C := \sqrt{r_2/r_1}\left((2M)^{1/p} + \sqrt{4c/r_1} + R_B\right) + R_B,$$

$$B := \left\{x \in \mathbb{R}^d : \|x - \mathbb{E}(X)\| \leq R_B\right\},$$

$$C := \left\{x \in \mathbb{R}^d : \|x - \mathbb{E}(X)\| \leq R_C\right\},$$

$$\tau := \min\left\{\sqrt{\frac{\epsilon}{2r_2}}, \frac{\epsilon}{2(R_B + R_C)r_2}\right\},$$

*and let $\mathcal{N}$ be a cover of $C$ by $\| \cdot \|$-balls with radius $\tau$; in the case that $\| \cdot \|$ is an $l_p$ norm, the size of this cover has bound*

$$|\mathcal{N}| \leq \left(1 + \frac{2R_C d}{\tau}\right)^d.$$

*Then with probability at least $1 - 3\delta$ over the draw of a sample of size $m \geq \max\{p'/(e2^{p'}\epsilon), 9\ln(1/\delta)\}$, every set of centers $P \in \mathcal{H}_f(\rho; c, k) \cup \mathcal{H}_f(\hat{\rho}; c, k)$ satisfies*

$$\left| \int \phi_f(x; P)d\rho(x) - \int \phi_f(x; P)d\hat{\rho}(x) \right| \leq 4\epsilon + 4r_2 R_C^2\sqrt{\frac{1}{2m}\ln\left(\frac{2|\mathcal{N}|^k}{\delta}\right)} + \sqrt{\frac{e2^{p'}\epsilon p'}{2m}}\left(\frac{2}{\delta}\right)^{1/p'}.$$

## 3.1 Compactification via Outer Brackets

The outer bracket is defined as follows.

**Definition 3.3.** *An outer bracket for probability measure $\nu$ at scale $\epsilon$ consists of two triples, one each for lower and upper bounds.*

1. *The function $\ell$, function class $Z_\ell$, and set $B_\ell$ satisfy two conditions: if $x \in B_\ell^c$ and $\phi \in Z_\ell$, then $\ell(x) \leq \phi(x)$, and secondly $|\int_{B_\ell^c} \ell(x) d\nu(x)| \leq \epsilon$.*

2. *Similarly, function $u$, function class $Z_u$, and set $B_u$ satisfy: if $x \in B_u^c$ and $\phi \in Z_u$, then $u(x) \geq \phi(x)$, and secondly $|\int_{B_u^c} u(x) d\nu(x)| \leq \epsilon$.*

Direct from the definition, given bracketing functions $(\ell, u)$, a bracketed function $\phi_f(\cdot; P)$, and the bracketing set $B := B_u \cup B_\ell$,

$$-\epsilon \leq \int_{B^c} \ell(x) d\nu(x) \leq \int_{B^c} \phi_f(x; P) d\nu(x) \leq \int_{B^c} u(x) d\nu(x) \leq \epsilon; \qquad (3.4)$$

in other words, as intended, this mechanism allows deviations on $B^c$ to be discarded. Thus to uniformly control the deviations of the dominated functions $Z := Z_u \cup Z_\ell$ over the set $B^c$, it suffices to simply control the deviations of the pair $(\ell, u)$.

The following lemma shows that a bracket exists for $\{\phi_f(\cdot; P) : P \in \mathcal{H}_f(\nu; c, k)\}$ and compact $B$, and moreover that this allows sampled points and candidate centers in far reaches to be deleted.

**Lemma 3.5.** *Consider the setting and definitions in Theorem 3.2, but additionally define*

$$M' := 2^{p'}\epsilon, \qquad \ell(x) := 0, \qquad u(x) := 4r_2 \|x - \mathbb{E}(X)\|^2, \qquad \epsilon_{\hat{\rho}} := \epsilon + \sqrt{\frac{M' e p'}{2m}} \left(\frac{2}{\delta}\right)^{1/p'}.$$

*The following statements hold with probability at least $1 - 2\delta$ over a draw of size $m \geq \max\{p'/(M'e), 9\ln(1/\delta)\}$.*

1. *$(u, \ell)$ is an outer bracket for $\rho$ at scale $\epsilon_\rho := \epsilon$ with sets $B_\ell = B_u = B$ and $Z_\ell = Z_u = \{\phi_f(\cdot; P) : P \in \mathcal{H}_f(\hat{\rho}; c, k) \cup \mathcal{H}_f(\rho; c, k)\}$, and furthermore the pair $(u, \ell)$ is also an outer bracket for $\hat{\rho}$ at scale $\epsilon_{\hat{\rho}}$ with the same sets.*

2. *For every $P \in \mathcal{H}_f(\hat{\rho}; c, k) \cup \mathcal{H}_f(\rho; c, k)$,*

$$\left| \int \phi_f(x; P) d\rho(x) - \int_B \phi_f(x; P \cap C) d\rho(x) \right| \leq \epsilon_\rho = \epsilon.$$

*and*

$$\left| \int \phi_f(x; P) d\hat{\rho}(x) - \int_B \phi_f(x; P \cap C) d\hat{\rho}(x) \right| \leq \epsilon_{\hat{\rho}}.$$

The proof of Lemma 3.5 has roughly the following outline.

1. Pick some ball $B_0$ which has probability mass at least $1/4$. It is not possible for an element of $\mathcal{H}_f(\hat{\rho}; c, k) \cup \mathcal{H}_f(\rho; c, k)$ to have all centers far from $B_0$, since otherwise the cost is larger than $c$. (Concretely, "far from" means at least $\sqrt{4c/r_1}$ away; note that this term appears in the definitions of $B$ and $C$ in Theorem 3.2.) Consequently, at least one center lies near to $B_0$; this reasoning was also the first step in the $k$-means consistency proof due to $k$-means Pollard [1].

2. It is now easy to dominate $P \in \mathcal{H}_f(\hat{\rho}; c, k) \cup \mathcal{H}_f(\rho; c, k)$ far away from $B_0$. In particular, choose any $p_0 \in B_0 \cap P$, which was guaranteed to exist in the preceding point; since $\min_{p \in P} \mathsf{B}_f(x, p) \leq \mathsf{B}_f(x, p_0)$ holds for all $x$, it suffices to dominate $p_0$. This domination proceeds exactly as discussed in the introduction; in fact, the factor 4 appeared there, and again appears in the $u$ here, for exactly the same reason. Once again, similar reasoning can be found in the proof by Pollard [1].

3. Satisfying the integral conditions over $\rho$ is easy: it suffices to make $B$ huge. To control the size of $B_0$, as well as the size of $B$, and moreover the deviations of the bracket over $B$, the moment tools from Appendix A are used.

Now turning consideration back to the proof of Theorem 3.2, the above bracketing allows the removal of points and centers outside of a compact set (in particular, the pair of compact sets $B$ and $C$, respectively). On the remaining truncated data and set of centers, any standard tool suffices; for mathematical convenience, and also to fit well with the use of norms in the definition of moments as well as the conditions on the convex function $f$ providing the divergence $\mathsf{B}_f$, norm structure used throughout the other properties, covering arguments are used here. (For details, please see Appendix B.)

# 4 Interlude: Refined Estimates via Clamping

So far, rates have been given that guarantee uniform convergence when the distribution has a few moments, and these rates improve with the availability of higher moments. These moment conditions, however, do not necessarily reflect any natural cluster structure in the source distribution. The purpose of this section is to propose and analyze another distributional property which is intended to capture cluster structure. To this end, consider the following definition.

**Definition 4.1.** *Real number $R$ and compact set $C$ are a* clamp *for probability measure $\nu$ and family of centers $Z$ and cost $\phi_f$ at scale $\epsilon > 0$ if every $P \in Z$ satisfies*

$$|\mathbb{E}_\nu(\phi_f(X; P)) - \mathbb{E}_\nu (\min \{\phi_f(X; P \cap C) , R\})| \leq \epsilon.$$

Note that this definition is similar to the second part of the outer bracket guarantee in Lemma 3.5, and, predictably enough, will soon lead to another deviation bound.

**Example 4.2.** If the distribution has bounded support, then choosing a clamping value $R$ and clamping set $C$ respectively slightly larger than the support size and set is sufficient: as was reasoned in the construction of outer brackets, if no centers are close to the support, then the cost is bad. Correspondingly, the clamped set of functions $Z$ should again be choices of centers whose cost is not too high.

For a more interesting example, suppose $\rho$ is supported on $k$ small balls of radius $R_1$, where the distance between their respective centers is some $R_2 \gg R_1$. Then by reasoning similar to the bounded case, all choices of centers achieving a good cost will place centers near to each ball, and thus the clamping value can be taken closer to $R_1$. ∎

Of course, the above gave the existence of clamps under favorable conditions. The following shows that outer brackets can be used to show the existence of clamps in general. In fact, the proof is very short, and follows the scheme laid out in the bounded example above: outer bracketing allows the restriction of consideration to a bounded set, and some algebra from there gives a conservative upper bound for the clamping value.

**Proposition 4.3.** *Suppose the setting and definitions of Lemma 3.5, and additionally define*

$$R := 2((2M)^{2/p} + R_B^2).$$

*Then $(C, R)$ is a clamp for measure $\rho$ and center $\mathcal{H}_f(\rho; c, k)$ at scale $\epsilon$, and with probability at least $1 - 3\delta$ over a draw of size $m \geq \max\{p'/(M'e), 9\ln(1/\delta)\}$, it is also a clamp for $\hat\rho$ and centers $\mathcal{H}_f(\hat\rho; c, k)$ at scale $\epsilon_{\hat\rho}$.*

The general guarantee using clamps is as follows. The proof is almost the same as for Theorem 3.2, but note that this statement is not used quite as readily, since it first requires the construction of clamps.

**Theorem 4.4.** *Fix a norm $\| \cdot \|$. Let $(R, C)$ be a clamp for probability measure $\rho$ and empirical counterpart $\hat\rho$ over some center class $Z$ and cost $\phi_f$ at respective scales $\epsilon_\rho$ and $\epsilon_{\hat\rho}$, where $f$ has corresponding convexity constants $r_1$ and $r_2$. Suppose $C$ is contained within a ball of radius $R_C$, let $\epsilon > 0$ be given, define scale parameter*

$$\tau := \min \left\{ \sqrt{\frac{\epsilon}{2r_2}}, \frac{r_1\epsilon}{2r_2 R_3} \right\},$$

*and let $\mathcal{N}$ be a cover of $C$ by $\| \cdot \|$-balls of radius $\tau$ (as per lemma B.4, if $\| \cdot \|$ is an $l_p$ norm, then $|\mathcal{N}| \leq (1 + (2R_C d)/\tau)^d$ suffices). Then with probability at least $1 - \delta$ over the draw of a sample of size $m \geq p'/(M'e)$, every set of centers $P \in Z$ satisfies*

$$\left| \int \phi_f(x; P)d\rho(x) - \int \phi_f(x; P)d\hat\rho(x) \right| \leq 2\epsilon + \epsilon_\rho + \epsilon_{\hat\rho} + R^2 \sqrt{\frac{1}{2m} \ln \left( \frac{2|\mathcal{N}|^k}{\delta} \right)}.$$

Before adjourning this section, note that clamps and outer brackets disagree on the treatment of the outer regions: the former replaces the cost there with the fixed value $R$, whereas the latter uses the value 0. On the technical side, this is necessitated by the covering argument used to produce the final theorem: if the clamping operation instead truncated beyond a ball of radius $R$ centered at each $p \in P$, then the deviations would be wild as these balls moved and suddenly switched the value at a point from 0 to something large. This is not a problem with outer bracketing, since the same points (namely $B^c$) are ignored by every set of centers.

# 5 Mixtures of Gaussians

Before turning to the deviation bound, it is a good place to discuss the condition $\sigma_1 I \preceq \Sigma \preceq \sigma_2 I$, which must be met by every covariance matrix of every constituent Gaussian in a mixture.

The lower bound $\sigma_1 I \preceq \Sigma$, as discussed previously, is fairly common in practice, arising either via a Bayesian prior, or by implementing EM with an explicit condition that covariance updates are discarded when the eigenvalues fall below some threshold. In the analysis here, this lower bound is used to rule out two kinds of bad behavior.

1. Given a budget of at least 2 Gaussians, and a sample of at least 2 distinct points, arbitrarily large likelihood may be achieved by devoting one Gaussian to one point, and shrinking its covariance. This issue destroys convergence properties of maximum likelihood, since the likelihood score may be arbitrarily large over every sample, but is finite for well-behaved distributions. The condition $\sigma_1 I \preceq \Sigma$ rules this out.

2. Another phenomenon is a "flat" Gaussian, meaning a Gaussian whose density is high along a lower dimensional manifold, but small elsewhere. Concretely, consider a Gaussian over $\mathbb{R}^2$ with covariance $\Sigma = \operatorname{diag}(\sigma, \sigma^{-1})$; as $\sigma$ decreases, the Gaussian has large density on a line, but low density elsewhere. This phenomenon is distinct from the preceding in that it does not produce arbitrarily large likelihood scores over finite samples. The condition $\sigma_1 I \preceq \Sigma$ rules this situation out as well.

In both the hard and soft clustering analyses here, a crucial early step allows the assertion that good scores in some region mean the relevant parameter is nearby. For the case of Gaussians, the condition $\sigma_1 I \preceq \Sigma$ makes this problem manageable, but there is still the possibility that some far away, fairly uniform Gaussian has reasonable density. This case is ruled out here via $\sigma_2 I \succeq \Sigma$.

**Theorem 5.1.** *Let probability measure $\rho$ be given with order-$p$ moment bound $M$ according to norm $\| \cdot \|_2$ where $p \geq 8$ is a positive multiple of 4, covariance bounds $0 < \sigma_1 \leq \sigma_2$ with $\sigma_1 \leq 1$ for simplicity, and real $c \leq 1/2$ be given. Then with probability at least $1 - 5\delta$ over the draw of a sample of size $m \geq \max\left\{(p/(2^{p/4+2}e))^2, 8\ln(1/\delta), d^2\ln(\pi\sigma_2)^2\ln(1/\delta)\right\}$, every set of Gaussian mixture parameters $(\alpha, \Theta) \in \mathcal{S}_{\mathrm{mog}}(\hat{\rho}; c, k, \sigma_1, \sigma_2) \cup \mathcal{S}_{\mathrm{mog}}(\rho; c, k, \sigma_1, \sigma_2)$ satisfies*

$$\left| \int \phi_g(x; (\alpha, \Theta)) d\rho(x) - \int \phi_g(x; (\alpha, \Theta)) d\hat{\rho}(x) \right|$$
$$= \mathcal{O}\left( m^{-1/2+3/p} \left( 1 + \sqrt{\ln(m) + \ln(1/\delta)} + (1/\delta)^{4/p} \right) \right),$$

*where the $\mathcal{O}(\cdot)$ drops numerical constants, polynomial terms depending on $c$, $M$, $d$, and $k$, $\sigma_2/\sigma_1$, and $\ln(\sigma_2/\sigma_1)$, but in particular has no sample-dependent quantities.*

The proof follows the scheme of the hard clustering analysis. One distinction is that the outer bracket now uses both components; the upper component is the log of the largest possible density — indeed, it is $\ln((2\pi\sigma_1)^{-d/2})$ — whereas the lower component is a function mimicking the log density of the steepest possible Gaussian — concretely, the lower bracket's definition contains the expression $\ln((2\pi\sigma_2)^{-d/2}) - 2\|x - \mathbb{E}_\rho(X)\|_2^2/\sigma_1$, which lacks the normalization of a proper Gaussian, highlighting the fact that bracketing elements need not be elements of the class. Superficially, a second distinction with the hard clustering case is that far away Gaussians can not be entirely ignored on local regions; the influence is limited, however, and the analysis proceeds similarly in each case.

**Acknowledgments**

The authors thank the NSF for supporting this work under grant IIS-1162581.

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
