[Supplementary Material]

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

# A  Moment Bounds

This section provides the basic probability controls resulting from moments. The material deals with the following slight generalization of the bounded moment definition from Section 2.

**Definition A.1.** *A function $\tau : \mathcal{X} \to \mathbb{R}^d$ has* order-$p$ moment bound $M$ *for probability measure $\rho$ with respect to norm $\|\cdot\|$ if $\mathbb{E}_\rho(\|\tau(X)\|^l) \leq M$ for all $1 \leq l \leq p$. (For convenience, measure $\rho$ and norm $\|\cdot\|$ will be often be implicit.)*

To connect this to the earlier definition, simply choose the map $\tau(x) := x - \mathbb{E}_\rho(X)$. As was the case in Section 2, this definition requires a uniform bound across all $l^{\text{th}}$ moments for $1 \leq l \leq p$. Of course, working with a probability measure implies these moments are all finite when just the $p^{\text{th}}$ moment is finite. The significance of working with a bound across all moments will be discussed again in the context of Lemma A.3 below.

The first result controls the measures of balls thanks to moments. This result is only stated for the source distribution $\rho$, but Hoeffding's inequality suffices to control $\hat{\rho}$.

**Lemma A.2.** *Suppose $\tau$ has order-$p$ moment bound $M$. Then for any $\epsilon > 0$,*

$$\Pr\left[\|\tau(X)\| \leq (M/\epsilon)^{1/p}\right] \geq 1 - \epsilon.$$

*Proof.* If $M = 0$, the result is immediate. Otherwise, when $M > 0$, for any $R > 0$, by Chebyshev's inequality,

$$\Pr\left[\|\tau(X)\| < R\right] = 1 - \Pr\left[\|\tau(X)\| \geq R\right] \geq 1 - \frac{\mathbb{E}\|\tau(X)\|^p}{R^p} \geq 1 - \frac{M}{R^p};$$

the result follows by choosing $R := (M/\epsilon)^{1/p}$. $\qquad\qquad\square$

The following fact will be the basic tool for controlling empirical averages via moments. Both the statement and proof are close to one by Tao [18, Equation 7], which rather than bounded moments uses boundedness (almost surely). As discussed previously, the term $1/\delta^{1/l}$ overtakes $\ln(1/\delta)$ when $l = \ln(1/\delta)/\ln(\ln(1/\delta))$.

For simplicity, this result is stated in terms of univariate random variables; to connect with the earlier development, the random variable $X$ will be substituted with the map $x \mapsto \|\tau(x)\|$.

**Lemma A.3.** *(Cf. Tao [18, Equation 7].) Let $m$ i.i.d. copies $\{X_i\}_{i=1}^m$ of a random variable $X$, even integer $p \geq 2$, real $M > 0$ with $\mathbb{E}(|X - \mathbb{E}(X)|^l) \leq M$ for $2 \leq l \leq p$, and $\epsilon > 0$ be given. If $m \geq p/(M\epsilon)$, then*

$$\Pr\left(\left|\frac{1}{n}\sum_i X_i - \mathbb{E}(X)\right| \geq \epsilon\right) \leq \frac{2}{(\epsilon\sqrt{m})^p}\left(\frac{Mpe}{2}\right)^{p/2}.$$

*In other words, with probability at least $1 - \delta$ over a draw of size $m \geq p/(M\epsilon)$,*

$$\left|\frac{1}{n}\sum_i X_i - \mathbb{E}(X)\right| \leq \sqrt{\frac{Mpe}{2m}}\left(\frac{2}{\delta}\right)^{1/p}.$$

*Proof.* Without loss of generality, suppose $\mathbb{E}(X_1) = 0$ (i.e., given $Y_1$ with $\mathbb{E}(Y_1) \neq 0$, work with $X_i := Y_i - \mathbb{E}(Y_1)$). By Chebyshev's inequality,

$$\Pr\left(\left|\frac{1}{m}\sum_i X_i\right| \geq \epsilon\right) \leq \frac{\mathbb{E}\left|\frac{1}{m}\sum_i X_i\right|^p}{\epsilon^p} = \frac{\mathbb{E}\left|\sum_i X_i\right|^p}{(m\epsilon)^p}. \qquad\qquad\text{(A.4)}$$

Recalling $p$ is even, consider the term

$$\mathbb{E}\left|\sum_i X_i\right|^p = \mathbb{E}\left(\sum_i X_i\right)^p = \sum_{i_1,i_2,\ldots,i_p \in [m]} \mathbb{E}\left(\prod_{j=1}^p X_{i_j}\right).$$

If some $i_j$ is equal to none of the others, then, by independence, a term $\mathbb{E}(X_{i_j}) = 0$ is introduced and the product vanishes; thus the product is nonzero when each $i_j$ has some copy $i_j = i_{j'}$, and thus there are at most $p/2$ distinct values amongst $\{i_j\}_{j=1}^p$. Each distinct value contributes a term $\mathbb{E}(X^l) \leq \mathbb{E}(|X|^l) \leq M$ for some $2 \leq l \leq p$, and thus

$$\mathbb{E}\left|\sum_i X_i\right|^p \leq \sum_{r=1}^{p/2} M^r N_r, \tag{A.5}$$

where $N_r$ is the number of ways to choose a multiset of size $p$ from $[m]$, subject to the constraint that each number appears at least twice, and at most $r$ distinct numbers appear. One way to over-count this is to first choose a subset of size $r$ from $m$, and then draw from it (with repetition) $p$ times:

$$N_r \leq \binom{m}{r} r^p \leq \frac{m^r r^p}{r!} \leq \frac{m^r r^p}{(r/e)^r} = (me)^r r^{p-r}.$$

Plugging this into eq. (A.5), and thereafter re-indexing with $r := p/2 - j$,

$$\mathbb{E}\left|\sum_i X_i\right|^p \leq \sum_{r=1}^{p/2} (Mme)^r r^{p-r} \leq \sum_{r=1}^{p/2} (Mme)^r (p/2)^{p-r}$$

$$\leq \sum_{j=0}^{p/2} (Mme)^{p/2-j}(p/2)^{p/2+j} \leq \left(\frac{Mmpe}{2}\right)^{p/2} \sum_{j=0}^{p/2} \left(\frac{p}{2Mme}\right)^j.$$

Since $p \leq Mme$,

$$\mathbb{E}\left|\sum_i X_i\right|^p \leq 2\left(\frac{Mmpe}{2}\right)^{p/2},$$

and the result follows by plugging this into eq. (A.4). $\qquad\square$

Thanks to Chebyshev's inequality, proving Lemma A.3 boils down to controlling $\mathbb{E}|\sum_i X_i|^p$, which here relied on a combinatorial scheme by Tao [18, Equation 7]. There is, however, another approach to controlling this quantity, namely Rosenthal inequalities, which write this $p^{\text{th}}$ moment of the sum in terms of the $2^{\text{nd}}$ and $p^{\text{th}}$ moments of individual random variables (general material on these bounds can be found in the book of Boucheron et al. [12, Section 15.4], however the specific form provided here is most easily presented by Pinelis and Utev [19]). While Rosenthal inequalities may seem a more elegant approach, they involve different constants, and thus the approach and bound here are followed instead to suggest further work on how to best control $\mathbb{E}|\sum_i X_i|^p$.

Returning to task, as was stated in the introduction, the dominated convergence theorem provides that $\int_{B_i} \|x\|_2^2 d\rho(x) \to \int \|x\|_2^2 d\rho(x)$ (assuming integrability of $x \mapsto \|x\|_2^2$), where the sequence of balls $\{B_i\}_{i=1}^\infty$ grow in radius without bound; moment bounds allow the rate of this process to be quantified as follows.

**Lemma A.6.** *Suppose $\tau$ has order-$p$ moment bound $M$, and let $0 < k < p$ be given. Then for any $\epsilon > 0$, the ball*

$$B := \left\{x \in \mathcal{X} : \|\tau(X)\| \leq (M/\epsilon)^{1/(p-k)}\right\}$$

*satisfies*

$$\int_{B^c} \|\tau(x)\|^k d\rho(x) \leq \epsilon.$$

*Proof.* Let the $B$ be given as specified; an application of Lemma A.2 with $\epsilon' := (\epsilon^p/M^k)^{1/(p-k)}$ yields

$$\int \mathbb{1}[x \in B^c] d\rho(x) = \Pr[\|\tau(x)\| > (M/\epsilon)^{1/(p-k)}] = \Pr[\|\tau(x)\| > (M/\epsilon')^{1/p}] \leq \epsilon'.$$

By Hölder's inequality with conjugate exponents $p/k$ and $p/(p-k)$ (where the condition $0 < k < p$ means each lies within $(1, \infty)$),

$$\int_{B^c} \|\tau(x)\|^k d\rho(x) = \int \|\tau(x)\|^k \mathbb{1}[x \in B^c] d\rho(x)$$

$$\leq \left( \int \|\tau(x)\|^{k(p/k)} d\rho(x) \right)^{k/p} \left( \int \mathbb{1}[x \in B^c]^{p/(p-k)} d\rho(x) \right)^{(p-k)/p}$$

$$\leq (M)^{k/p} \left( \frac{\epsilon^{p/(p-k)}}{M^{k/(p-k)}} \right)^{(p-k)/p}$$

$$= \epsilon$$

as desired. $\qquad\square$

Lastly, thanks to the moment-based deviation inequality in Lemma A.3, the deviations on this outer region may be controlled. Note that in order to control the $k$-means cost (i.e., an exponent $k = 2$), at least 4 moments are necessary ($p \geq 4$).

**Lemma A.7.** *Let integers $k \geq 1$ and $p' \geq 1$ be given, and set $\tilde{p} := k(p'+1)$. Suppose $\tau$ has order-$\tilde{p}$ moment bound $M$, and let $\epsilon > 0$ be arbitrary. Define the radius $R$ and ball $B$ as*

$$R := \max\{(M/\epsilon)^{1/(\tilde{p}-ik)} : 1 \leq i < \tilde{p}/k\} \qquad \text{and} \qquad B := \{x \in \mathcal{X} : \|\tau(x)\| \leq R\},$$

*and set $M' := 2^{p'} \epsilon$. With probability at least $1 - \delta$ over the draw of a sample of size $m \geq p'/(M'e)$,*

$$\left| \int_{B^c} \|\tau(x)\|^k d\hat{\rho}(x) - \int_{B^c} \|\tau(x)\|^k d\rho(x) \right| \leq \sqrt{\frac{M'ep'}{2m}} \left( \frac{2}{\delta} \right)^{1/p'}.$$

*Proof.* Consider a fixed $1 \leq i < \tilde{p}/k = p' + 1$, and set $l = ik$. Let $B_l$ be the ball provided by Lemma A.6 for exponent $l$. Since $B \supseteq B_l$,

$$\int_{B^c} \|\tau(x)\|^l d\rho(x) \leq \int_{B_l^c} \|\tau(x)\|^l d\rho(x) \leq \epsilon.$$

As such, by Minkowski's inequality, since $z \mapsto z^l$ is convex for $l \geq 1$,

$$\left( \int \left| \|\tau(x)\| \mathbb{1}[x \in B^c] - \int_{B^c} \|\tau(x)\| d\rho(x) \right|^l d\rho(x) \right)^{1/l}$$

$$\leq \left( \int_{B^c} \|\tau(x)\|^l d\rho(x) \right)^{1/l} + \left( \int_{B^c} \|\tau(x)\| d\rho(x) \right)^{l/l}$$

$$\leq 2 \left( \int_{B^c} \|\tau(x)\|^l d\rho(x) \right)^{1/l},$$

meaning

$$\int \left| \|\tau(x)\| \mathbb{1}[x \in B^c] - \int_{B^c} \|\tau(x)\| d\rho(x) \right|^l d\rho(x) \leq 2^l \int_{B^c} \|\tau(x)\|^l \leq 2^l \int_{B_l^c} \|\tau(x)\|^l \leq 2^l \epsilon.$$

Since $l = ik$ had $1 \leq i < \tilde{p}/k = p' + 1$ arbitrary, it follows that the map $x \mapsto \|\tau(x)\|^k \mathbb{1}[x \in B^c]$ has its first $p'$ moments bounded by $2^{p'} \epsilon$.

The finite sample bounds will now proceed with an application of Lemma A.3, where the random variable $X$ will be the map $x \mapsto \|\tau(x)\|^k \mathbb{1}[x \in B^c]$. Plugging the above moment bounds for this random variable into Lemma A.3, the result follows. $\qquad\square$

# B  Deferred Material from Section 3

Before proceeding with the main proofs, note that Bregman divergences in the setting here are sandwiched between quadratics.

**Lemma B.1.** *If differentiable $f$ is $r_1$ strongly convex with respect to $\|\cdot\|$, then $\mathsf{B}_f(x,y) \geq r_1\|x-y\|^2$. If differentiable $f$ has Lipschitz gradients with parameter $r_2$ with respect to $\|\cdot\|$, then $\mathsf{B}_f(x,y) \leq r_2\|x-y\|^2$.*

*Proof.* The first part (strong convexity) is standard (see for instance the proof by Shalev-Shwartz [20, Lemma 13], or a similar proof by Hiriart-Urruty and Lemaréchal [21, Theorem B.4.1.4]). For the second part, by the fundamental theorem of calculus, properties of norm duality, and the Lipschitz gradient property,

$$f(x) = f(y) + \langle \nabla f(y), x-y \rangle + \int_0^1 \langle \nabla f(y + t(x-y)) - \nabla f(y), x - y \rangle \, dt$$

$$\leq f(y) + \langle \nabla f(y), x-y \rangle + \int_0^1 \|\nabla f(y + t(x-y)) - \nabla f(y)\|_* \|x - y\| \, dt$$

$$\leq f(y) + \langle \nabla f(y), x-y \rangle + \frac{r_2}{2}\|x - y\|^2.$$

(The preceding is also standard; see for instance the beginning of a proof by Hiriart-Urruty and Lemaréchal [21, Theorem E.4.2.2], which only differs by fixing the norm $\|\cdot\|_2$.) $\qquad\square$

## B.1  Proof of Lemma 3.5

The first step is the following characterization of $\mathcal{H}_f(\nu; c, k)$: at least one center must fall within some compact set. (The lemma works more naturally with the contrapositive.) The proof by Pollard [1] also started by controlling a single center.

**Lemma B.2.** *Consider the setting of Lemma 3.5, and additionally define the two balls*

$$B_0 := \left\{ x \in \mathbb{R}^d : \|x - \mathbb{E}_\rho(X)\| \leq (2M)^{1/p} \right\},$$

$$C_0 := \left\{ x \in \mathbb{R}^d : \|x - \mathbb{E}_\rho(X)\| \leq (2M)^{1/p} + \sqrt{4c/r_1} \right\},$$

*Then $\rho(B_0) \geq 1/2$, and for any center set $P$, if $P \cap C_0 = \emptyset$ then $\mathbb{E}_\rho(\phi_f(X; P)) \geq 2c$. Furthermore, with probability at least $1 - \delta$ over a draw from $\rho$ of size at least*

$$m \geq 9 \ln\left(\frac{1}{\delta}\right).$$

*then $\hat{\rho}(B_0) > 1/4$ and $P \cap C_0 = \emptyset$ implies $\mathbb{E}_{\hat{\rho}}(\phi_f(X; P)) > c$.*

*Proof.* The guarantee $\rho(B_0) \geq 1/2$ is direct from Lemma A.2 with moment map $\tau(x) := x - \mathbb{E}_\rho(X)$. By Hoeffding's inequality and the lower bound on $m$, with probability at least $1 - \delta$,

$$\hat{\rho}(B_0) \geq \rho(B_0) - \sqrt{\frac{1}{2m} \ln\left(\frac{1}{\delta}\right)} > \frac{1}{4}.$$

By the definition of $C_0$, every $p \in C_0^c$ and $x \in B_0$ satisfies

$$\mathsf{B}_f(x, p) \geq r_1\|x - p\|^2 \geq 4c.$$

Now let $\nu$ denote either $\rho$ or $\hat{\rho}$; then for any set of centers $P$ with $P \cap C_0 = \emptyset$ (meaning $P \subseteq C_0^c$),

$$\int \phi_f(x; P) d\nu(x) = \int \min_{p \in P} \mathsf{B}_f(x, p) d\nu(x)$$

$$\geq \int_{B_0} \min_{p \in P} \mathsf{B}_f(x, p) d\nu(x)$$

$$\geq \int_{B_0} \min_{p \in P} 4c \, d\nu(x)$$

$$= 4c\nu(B_0).$$

Instantiating $\nu$ with $\rho$ or $\hat{\rho}$, the results follow. $\qquad\square$

With this tiny handle on the structure of a set of centers $P$ satisfying $\phi_f(x; P) \leq c$, the proof of Lemma 3.5 follows.

*Proof of Lemma 3.5.* Throughout both sections, let $B_0$ and $C_0$ be as defined in Lemma B.2; it follows by Lemma B.2, with probability at least $1 - \delta$, that $P \in \mathcal{H}_f(\rho; c, k) \cup \mathcal{H}_f(\hat{\rho}; c, k)$ implies $P \cap C_0 \neq \emptyset$. Henceforth discard this failure event, and fix any $P \in \mathcal{H}_f(\rho; c, k) \cup \mathcal{H}_f(\hat{\rho}; c, k)$.

1. Since $P \cap C_0 \neq \emptyset$, fix some $p_0 \in P \cap C_0$. Since $B \supseteq C_0$ by definition, it follows, for every $x \in B^c$ that

$$\phi_f(x; P) = \min_{p \in P} \mathsf{B}_f(x, p) \leq r_2 \|x - p_0\|^2 \leq r_2(\|x - \mathbb{E}_\rho(X)\| + \|p_0 - \mathbb{E}_\rho(X)\|)^2$$

$$\leq 4r_2 \|x - \mathbb{E}_\rho(X)\|^2 = u(x).$$

   Additionally,

$$\ell(x) = 0 \leq \min_{p \in P} r_1 \|x - p\|^2 \leq \phi_f(x; P),$$

   meaning $u$ and $\ell$ properly bracket $Z_\ell = Z_u$ over $B^c$; what remains is to control their mass over $B^c$.

   Since $\ell = 0$,

$$\left| \int_{B^c} \ell(x) d\hat{\rho}(x) \right| = \left| \int_{B^c} \ell(x) d\rho(x) \right| = 0 < \epsilon.$$

   Next, for $u$ with respect to $\rho$, the result follows from the definition of $u$ together with Lemma A.6 (using the map $\tau(x) = x - \mathbb{E}_\rho(X)$ together with exponent 2).

   Lastly, to control $u$ with respect to $\hat{\rho}$, note that $p' \leq p/2 - 1$ means $\tilde{p} := 2(p'+1) \leq p$, and thus the map $\tau(x) := \|x - \mathbb{E}_\rho(X)\|^2$ has order-$\tilde{p}$ moment bound $M$. Thus, by Lemma A.7 and the triangle inequality,

$$\left| \int_{B^c} u(x) d\hat{\rho}(x) \right| \leq \epsilon + \sqrt{\frac{M'ep'}{2m}} \left( \frac{2}{\delta} \right)^{1/p'} = \epsilon_{\hat{\rho}}.$$

2. Throughout this proof, let $\nu$ denote either $\rho$ or $\hat{\rho}$; the above established

$$\left| \int_{B^c} u(x) d\nu(x) \right| \leq \epsilon_\nu,$$

   where in the case of $\nu = \hat{\rho}$, this statement holds with probability $1 - \delta$; henceforth discard this failure event, and thus the statement holds for both cases.

   By definition of $C$, for any $p \in C^c$ and $x \in B$,

$$\mathsf{B}_f(x, p) \geq r_1 \|x - p\|^2 \geq r_1 \left( \sqrt{r_2/r_1} \left( (2M)^{1/p} + \sqrt{4c/r_1} + R_B \right) \right)^2$$

$$= r_2 \left( (2M)^{1/p} + \sqrt{4c/r_1} + R_B \right)^2.$$

   On the other hand, fixing any $p_0 \in P \cap C_0$ (which was guaranteed to exist at the start of this proof), since $C_0 \subseteq C$,

$$\sup_{x \in B} \phi_f(x; P \cap C) \leq \sup_{x \in B} r_2 \|x - p_0\|^2 \leq r_2 \left( (2M)^{1/p} + \sqrt{4c/r_1} + R_B \right)^2.$$

   Consequently, no element of $B$ is closer to an element of $P \cap C$ than to any element of $P \setminus C$. As such,

$$\int \phi_f(x; P) d\nu(x) \geq \int_B \phi_f(x; P) d\nu(x) + \int_{B^c} \ell(x) d\nu(x) = \int_B \phi_f(x; P \cap C) d\nu(x).$$

   (Note here that $\ell(x) = 0$ was used directly, rather than the $\epsilon$ provided by outer covering; in the case of Gaussian mixtures, both bracket elements are nonzero, and $\epsilon$ will be used.) This establishes one direction of the bound.

For the other direction, note that adding centers back in only decreases cost (because $\min_{p \in P \cap C}$ is replaced with $\min_{p \in P}$), and thus recalling the properties of the outer bracket element $u$ established above,

$$
\begin{aligned}
\int_B \phi_f(x; P \cap C) d\nu(x) &= \int \phi_f(x; P \cap C) d\nu(x) - \int_{B^c} \phi_f(x; P \cap C) d\nu(x) \\
&\geq \int \phi_f(x; P \cap C) d\nu(x) - \int_{B^c} u(x) d\nu(x) \\
&\geq \int \phi_f(x; P) d\nu(x) - \epsilon_\nu,
\end{aligned}
$$

which gives the result(s).

$\square$

## B.2 Covering Properties

The next step is to control the deviations over the bounded portion; this is achieved via uniform covers, as developed in this subsection.

First, another basic fact about Bregman divergences.

**Lemma B.3.** *Let differentiable convex function $f$ be given with Lipschitz gradient constant $r_2$ with respect to norm $\| \cdot \|$, and let $\mathsf{B}_f$ be the corresponding Bregman divergence. For any $\{x, y, z\} \subseteq \mathcal{X}$,*

$$
\mathsf{B}_f(x, z) \leq \mathsf{B}_f(x, y) + \mathsf{B}_f(y, z) + r_2 \|x - y\| \|y - z\|.
$$

*Similarly, given finite sets $Y \subseteq \mathcal{X}$ and $Z \subseteq \mathcal{X}$, and letting $Y(p)$ and $Z(p)$ respectively select (any) closest point in $Y$ and $Z$ to $p$ according to $\mathsf{B}_f$, meaning*

$$
Y(p) := \arg\min_{y \in Y} \mathsf{B}_f(y, p) \qquad \text{and} \qquad Z(p) := \arg\min_{z \in Z} \mathsf{B}_f(z, p),
$$

*then*

$$
\min_{z \in Z} \mathsf{B}_f(x, z) \leq \min_{y \in Y} \mathsf{B}_f(x, y) + \mathsf{B}_f(Y(x), Z(Y(x))) + r_2 \|x - Y(x)\| \|Y(x) - Z(Y(x))\|.
$$

*Proof.* By definition of $\mathsf{B}_f$, properties of dual norms, and the Lipschitz gradient property,

$$
\begin{aligned}
\mathsf{B}_f(x, z) - \mathsf{B}_f(x, y) - \mathsf{B}(y, z) &= f(x) - f(z) - f(x) + f(y) - f(y) + f(z) \\
&\quad - \langle \nabla f(z), x - z \rangle + \langle \nabla f(y), x - y \rangle + \langle \nabla f(z), y - z \rangle \\
&= \langle \nabla f(y) - \nabla f(z), x - y \rangle \\
&\leq \|\nabla f(y) - \nabla f(z)\|_* \|x - y\| \\
&\leq r_2 \|y - z\| \|x - y\|;
\end{aligned}
$$

rearranging this inequality gives the first statement.

The second statement follows the first instantiated with $y = Y(x)$ and $z = Z(Y(x))$, since

$$
\begin{aligned}
\min_{z \in Z} \mathsf{B}_f(x, z) &\leq \mathsf{B}_f(x, Z(Y(x))) \\
&\leq \mathsf{B}_f(x, Y(x)) + \mathsf{B}_f(Y(x), Z(Y(x))) + r_2 \|x - Y(x)\| \|Y(x) - Z(Y(x))\|,
\end{aligned}
$$

and using $\mathsf{B}_f(x, Y(x)) = \min_{y \in Y} \mathsf{B}_f(x, y)$. $\square$

The covers will be based on norm balls; the following estimate is useful.

**Lemma B.4.** *If $\| \cdot \|$ is an $l_p$ norm over $\mathbb{R}^d$, then the ball of radius $R$ admits a cover $\mathcal{N}$ with size*

$$
|\mathcal{N}| \leq \left(1 + \frac{2Rd}{\tau}\right)^d.
$$

*Proof.* It suffices to grid the $B$ with $l_\infty$ balls centered at grid points at scale $\tau/d$; the result follows since the $l_\infty$ balls of radius $\tau/d$ are contained in $l_p$ balls of radius $\tau$ for all $p \geq 1$. $\square$

The uniform covering result is as follows.

**Lemma B.5.** *Let scale $\epsilon > 0$, ball $B := \{x \in \mathbb{R}^d : \|x - \mathbb{E}(X)\| \leq R\}$, parameter set $Z := \{x \in \mathbb{R}^d : \|x - \mathbb{E}(X)\| \leq R_2\}$, and differentiable convex function $f$ with Lipschitz gradient parameter $r_2$ with respect to norm $\|\cdot\|$ be given. Define resolution parameter*

$$\tau := \min\left\{ \sqrt{\frac{\epsilon}{2r_2}}, \frac{\epsilon}{2(R_2 + R)r_2} \right\},$$

*and let $\mathcal{N}$ be set of centers for a cover of $Z$ by $\|\cdot\|$-balls of radius $\tau$ (see Lemma B.4 for an estimate when $\|\cdot\|$ is an $l_p$ norm). It follows that there exists a uniform cover $\mathcal{F}$ at scale $\epsilon$ with cardinality $|\mathcal{N}|^k$, meaning for any collection $P = \{p_i\}_{i=1}^l$ with $p_i \in Z$ and $l \leq k$, there is a cover element $Q$ with*

$$\sup_{x \in B} \left| \min_{p \in P} \mathsf{B}_f(x, p) - \min_{q \in Q} \mathsf{B}_f(x, q) \right| \leq \epsilon.$$

*Proof.* Given a collection $P$ as specified, choose $Q$ so that for every $p \in P$, there is $q \in Q$ with $\|p - q\| \leq \tau$, and vice versa. By Lemma B.3 (and using the notation therein), for any $x \in B^c$,

$$\min_{p \in P} \mathsf{B}_f(x, p) \leq \min_{q \in Q} \mathsf{B}_f(x, q) + \mathsf{B}_f(Q(x), P(Q(x))) + r_2 \|x - Q(x)\| \|Q(x) - P(Q(x))\|$$

$$\leq \min_{q \in Q} \mathsf{B}_f(x, q) + r_2 \tau^2 + r_2 \tau (R + R_2)$$

$$\leq \min_{q \in Q} \mathsf{B}_f(x, q) + \epsilon;$$

the reverse inequality holds for the same reason, and the result follows. $\square$

## B.3 Proof of Theorem 3.2 and Corollary 3.1

First, the proof of the general rate for $\mathcal{H}_f(\nu; c, k)$.

*Proof of Theorem 3.2.* For convenience, define $M' = 2^{p'}\epsilon$. By Lemma B.5, let $\mathcal{N}$ be a cover of the set $C$, whereby every set of centers $P \subseteq C$ with $|P| \leq k$ has a cover element $Q \in \mathcal{N}^k$ with

$$\sup_{x \in B} \left| \min_{p \in P} \mathsf{B}_f(x, p) - \min_{q \in Q} \mathsf{B}_f(x, q) \right| \leq \epsilon; \tag{B.6}$$

when $\|\cdot\|$ is an $l_p$ norm, Lemma B.4 provides the stated estimate of its size. Since $B \subseteq C$ and

$$\sup_{x \in B} \sup_{p \in C} \mathsf{B}_f(x, p) \leq r_2 \sup_{x \in B} \sup_{p \in C} \|x - p\|^2 \leq 4r_2 R_C^2,$$

it follows by Hoeffding's inequality and a union bound over $\mathcal{N}^k$ that with probability at least $1 - \delta$,

$$\sup_{Q \in \mathcal{N}^k} \left| \int_B \phi(x; Q) d\hat{\rho}(x) - \int_B \phi(x; Q) d\rho(x) \right| \leq 4r_2 R_C^2 \sqrt{\frac{1}{2m} \ln\left( \frac{2|\mathcal{N}|^k}{\delta} \right)}. \tag{B.7}$$

For the remainder of this proof, discard the corresponding failure event.

Now let any $P \in \mathcal{H}_f(\rho; c, k) \cup \mathcal{H}_f(\hat{\rho}; c, k)$ be given, and let $Q \in \mathcal{N}^k$ be a cover element satisfying eq. (B.6) for $P \cap C$. By eq. (B.6), eq. (B.7), and Lemma 3.5 (and thus discarding an additional

failure event having probability $2\delta$),

$$
\left| \int \phi_f(x; P)d\rho(x) - \int \phi_f(x; P)d\hat{\rho}(x) \right| \leq \left| \int \phi_f(x; P)d\rho(x) - \int_B \phi_f(x; P \cap C)d\rho(x) \right|
$$
$$
+ \left| \int_B \phi_f(x; P \cap C)d\rho(x) - \int_B \phi_f(x; Q)d\rho(x) \right|
$$
$$
+ \left| \int_B \phi_f(x; Q)d\rho(x) - \int_B \phi_f(x; Q)d\hat{\rho}(x) \right|
$$
$$
+ \left| \int_B \phi_f(x; Q)d\hat{\rho}(x) - \int_B \phi_f(x; P \cap C)d\hat{\rho}(x) \right|
$$
$$
+ \left| \int_B \phi_f(x; P \cap C)d\hat{\rho}(x) - \int \phi_f(x; P)d\hat{\rho}(x) \right|
$$
$$
\leq 2\epsilon + 4r_2 R_C^2 \sqrt{\frac{1}{2m} \ln\left(\frac{2|\mathcal{N}|^k}{\delta}\right)} + \epsilon_\rho + \epsilon_{\hat{\rho}},
$$

and the result follows by unwrapping the definitions of $\epsilon_\rho$ and $\epsilon_{\hat{\rho}}$ from Lemma 3.5, and $M' = 2^{p'}\epsilon$ as above. $\qquad\square$

The more concrete bound for the $k$-means cost is proved as follows.

*Proof of Corollary 3.1.* Set

$$
\epsilon := m^{-1/2+1/p}, \qquad p' := p/4, \qquad M' := 2^{p'}\epsilon = 2^{p/4}m^{-1/2+1/p},
$$

and recall $f(x) := \|x\|_2^2$ has convexity constants $r_1 = r_2 = 2$. Since

$$
m = \sqrt{m}\sqrt{m} \geq \frac{p\sqrt{m}}{2^{p/4+2}e} \geq \frac{p'm^{1/2-1/p}}{2^{p'}e} = \frac{p'}{M'e}
$$

and $p' = p/2 - p/4 \leq p/2 - 1$, the conditions for Theorem 3.2 are met, and thus with probability at least $1 - \delta$,

$$
\left| \int \phi_f(x; P)d\rho(x) - \int \phi_f(x; P)d\hat{\rho}(x) \right| \leq 4\epsilon + 4R_C^2\sqrt{\frac{1}{2m}\ln\left(\frac{2|\mathcal{N}|^k}{\delta}\right)} + \sqrt{\frac{2^{p/4}ep\epsilon}{8m}}\left(\frac{2}{\delta}\right)^{4/p},
$$

where

$$
R_C := (2M)^{1/p} + \sqrt{2c} + 2R_B,
$$
$$
R_B := \max\left\{ (2M)^{1/p} + \sqrt{2c}, \max_{i \in [p']}(M/\epsilon)^{1/(p-2i)} \right\},
$$
$$
|\mathcal{N}| \leq \left(1 + \frac{2R_C d}{\tau}\right)^d,
$$
$$
\tau := \min\left\{ \sqrt{\frac{\epsilon}{4}}, \frac{\epsilon}{4(R_B + R_C)} \right\}.
$$

To simplify these quantities, since $\epsilon \leq 1$, the term $1/\epsilon^{1/(p-2i)}$, as $i$ ranges between 1 and $p - 2p'$, is maximized at $i = 1/(p - 2p') = 2/p$. Therefore, by choice of $M_1$ and $\epsilon$,

$$
R_B \leq c_1 + (M/\epsilon)^{1/(p-2)} + (M/\epsilon)^{1/(p-2p')} \leq c_1 + (M^{1/(p-2)} + M^{1/(p-2p')})/\epsilon^{2/p}
$$
$$
= c_1 + M_1 m^{1/p-2/p^2}.
$$

Consequently,

$$
R_C = c_1 + 2R_B \leq 3c_1 + 2M_1 m^{1/p-2/p^2} \qquad \text{and} \qquad R_C^2 \leq 18c_1^2 + 8M_1^2 m^{2/p-4/p^2}.
$$

This entails

$$\frac{2R_C d}{\tau} \leq 2R_C d\left(2m^{1/4-1/(2p)} + 4(R_B + R_C)m^{1/2-1/p}\right)$$

$$\leq 8d\left((3c_1 + 2M_1 m^{1/p-2/p^2})m^{1/4-1/(2p)} + (36c_1^2 + 16M_1^2 m^{2/p-4/p^2})m^{1/2-1/p}\right)$$

$$\leq 288dm(c_1 + c_1^2 + M_1 + M_1^2).$$

Secondly,

$$\frac{R_C^2}{\sqrt{m}} \leq (18c_1^2 + 8M_1^2 m^{2/p-4/p^2})m^{-1/2} \leq m^{\min\{1/4,-1/2+2/p\}}(18c_1^2 + 8M_1^2).$$

The last term is direct, since

$$\sqrt{\epsilon/m} = m^{-1/4+1/(2p)-1/2} = m^{-1/2+1/(2p)}m^{-1/4}.$$

Combining these pieces, the result follows. □

## C   Deferred Material from Section 4

First, the deferred proof that outer brackets give rise to clamps.

*Proof of Proposition 4.3.* Throughout this proof, let $\nu$ refer to either $\rho$ or $\hat{\rho}$, with $\epsilon_\nu$ similarly referring to either $\epsilon_\rho$ or $\epsilon_{\hat{\rho}}$. Let $P \in \mathcal{H}_f(\rho; c, k) \cup \mathcal{H}_f(\hat{\rho}; c, k)$ be given.

One direction is direct:

$$\int \phi_f(x; P)d\nu(x) \geq \int \phi_f(x; P \cap C)d\nu(x)$$

$$\geq \int \min\{\phi_f(x; P \cap C), R\}d\nu(x).$$

For the second direction, with probability at least $1 - \delta$, Lemma B.2 grants the existence of $p' \in P \cap C_0 \subseteq P \cap C$. Consequently, for any $x \in B$,

$$\min_{p \in P} \mathsf{B}_f(x, p) \leq \min_{p \in P \cap C} \mathsf{B}_f(x, p) \leq \mathsf{B}_f(x, p')$$

$$\leq r_2\|x - p'\|^2 \leq 2r_2\left(\|x - \mathbb{E}_\rho(X)\|^2 + \|p' - \mathbb{E}_\rho(X)\|^2\right)$$

$$\leq R;$$

in other words, if $x \in B$, then $\min\{\phi_f(x; P \cap C), R\} = \phi_f(x; P \cap C)$. Combining this with the last part of Lemma 3.5.

$$\int \min\{\phi_f(x; P \cap C), R\}d\nu(x) \geq \int_B \min\{\phi_f(x; P \cap C), R\}d\nu(x)$$

$$\geq \int_B \phi_f(x; P \cap C)d\nu(x)$$

$$\geq \int \phi_f(x; P)d\nu(x) - \epsilon_\nu.$$

□

The proof of Theorem 4.4 will depend on the following uniform covering property of the clamped cost (which mirrors Lemma B.5 for the unclamped cost).

**Lemma C.1.** *Let scale $\epsilon > 0$, clamping value $R_3$, parameter set $C$ contained within a $\|\cdot\|$-ball of some radius $R_2$, and differentiable convex function $f$ with Lipschitz gradient parameter $r_2$ and strong convexity modulus $r_1$ with respect to norm $\|\cdot\|$ be given. Define resolution parameter*

$$\tau := \min\left\{\sqrt{\frac{\epsilon}{2r_2}}, \frac{r_1\epsilon}{2r_2 R_3}\right\},$$

and let $\mathcal{N}$ be set of centers for a cover of $C$ by $\|\cdot\|$-balls of radius $\tau$ (see Lemma B.4 for an estimate when $\|\cdot\|$ is an $l_p$ norm). It follows that there exists a uniform cover $\mathcal{F}$ at scale $\epsilon$ with cardinality $|\mathcal{N}|^k$, meaning for any collection $P = \{p_i\}_{i=1}^l$ with $p_i \in C$ and $l \leq k$, there is a cover element $Q$ with

$$\sup_x \left| \min \left\{ R_3, \min_{p \in P} \mathsf{B}_f(x, p) \right\} - \min \left\{ R_3, \min_{q \in Q} \mathsf{B}_f(x, q) \right\} \right| \leq \epsilon.$$

*Proof.* Given a collection $P$ as specified, choose $Q$ so that for every $p \in P$, there is $q \in Q$ with $\|p - q\| \leq \tau$, and vice versa.

First suppose $\min_{q \in Q} \mathsf{B}_f(x, q) \geq R_3$; then

$$\min \left\{ R_3, \min_{p \in P} \mathsf{B}_f(x, p) \right\} \leq R_3 = \min \left\{ R_3, \min_{q \in Q} \mathsf{B}_f(x, q) \right\}$$

as desired.

Otherwise, $\min_{q \in Q} \mathsf{B}_f(x, q) < R_3$, which by the sandwiching property (cf. Lemma B.1) means

$$r_1 \|x - Q(x)\| \leq \mathsf{B}_f(x, Q(x)) < R_3.$$

By Lemma B.3,

$$\min \left\{ R_3, \min_{p \in P} \mathsf{B}_f(x, p) \right\} \leq \min \left\{ R_3, \min_{q \in Q} \mathsf{B}_f(x, q) + \mathsf{B}_f(Q(x), P(Q(x))) + r_2 \|x - Q(x)\| \|Q(x) - P(Q(x))\| \right\}$$

$$\leq \min \left\{ R_3, \min_{q \in Q} \mathsf{B}_f(x, q) + r_2 \tau^2 + r_2 \tau \|x - Q(x)\| \right\}$$

$$\leq \min \left\{ R_3, \min_{q \in Q} \mathsf{B}_f(x, q) + r_2 \tau^2 + \frac{r_2 R_3}{r_1} \tau \right\}$$

$$\leq \min \left\{ R_3, \min_{q \in Q} \mathsf{B}_f(x, q) \right\} + \epsilon.$$

The reverse inequality is analogous. □

The proof of Theorem 4.4 follows.

*Proof of Theorem 4.4.* This proof is a minor alteration of the proof of Theorem 3.2.

By Lemma C.1, let $\mathcal{N}$ be a cover of the set $C$, whereby every set of centers $P \subseteq C$ with $|P| \leq k$ has a cover element $Q \in \mathcal{N}^k$ with

$$\sup_x |\min\{\phi_f(x; P), R\} - \min\{\phi_f(x; Q), R\}| \leq \epsilon; \tag{C.2}$$

when $\|\cdot\|$ is an $l_p$ norm, Lemma B.4 provides the stated estimate of its size. Since $\min\{\phi_f(x; Q), R\} \in [0, R]$, it follows by Hoeffding's inequality and a union bound over $\mathcal{N}^k$ that with probability at least $1 - \delta$,

$$\sup_{Q \in \mathcal{N}^k} \left| \int_B \phi_f(x; Q) d\hat{\rho}(x) - \int_B \phi_f(x; Q) d\rho(x) \right| \leq R \sqrt{\frac{1}{2m} \ln \left( \frac{2|\mathcal{N}|^k}{\delta} \right)}. \tag{C.3}$$

For the remainder of this proof, discard the corresponding failure event.

Now let any $P \in Z$ be given, and let $Q \in \mathcal{N}^k$ be a cover element satisfying eq. (C.2) for $P \cap C$. By eq. (C.2), eq. (C.3), and lastly by the definition of clamp,

$$\left| \int \phi_f(x; P) d\rho(x) - \int \phi_f(x; P) d\hat{\rho}(x) \right| \leq \left| \int \phi_f(x; P) d\rho(x) - \int \min\{\phi_f(x; P \cap C), R\} d\rho(x) \right|$$

$$+ \left| \int \min\{\phi_f(x; P \cap C), R\} d\rho(x) - \int \min\{\phi_f(x; Q), R\} d\rho(x) \right|$$

$$+ \left| \int \min\{\phi_f(x; Q), R\} d\rho(x) - \int \min\{\phi_f(x; Q), R\} d\hat{\rho}(x) \right|$$

$$+ \left| \int \min\{\phi_f(x; Q), R\} d\hat{\rho}(x) - \int \min\{\phi_f(x; P \cap C), R\} d\hat{\rho}(x) \right|$$

$$+ \left| \int \min\{\phi_f(x; P \cap C), R\} d\hat{\rho}(x) - \int \phi_f(x; P) d\hat{\rho}(x) \right|$$

$$\leq 2\epsilon + \epsilon_\rho + \epsilon_{\hat{\rho}} + R^2 \sqrt{\frac{1}{2m} \ln \left( \frac{2|\mathcal{N}|^k}{\delta} \right)}.$$

$\square$

## D  Deferred Material from Section 5

The following notation for restricting a Gaussian mixture to a certain set of means will be convenient throughout this section.

**Definition D.1.** *Given a Gaussian mixture with parameters* $(\alpha, \Theta)$ *(where* $\alpha = \{\alpha_i\}_{i=1}^k$ *and* $\Theta = \{\theta_i\}_{i=1}^k = \{(\mu_i, \Sigma_i)\}_{i=1}^k$*), and a set of means* $B \subseteq \mathbb{R}^d$*, define*

$$(\alpha, \Theta) \sqcap B := \{(\{\alpha_i\}_{i \in I}, \{(\mu_i, \Sigma_i)\}_{i \in I}) : I = \{1 \leq i \leq k : \mu_i \in B\}\}.$$

*(Note that potentially* $\sum_{i \in I} \alpha_i < 1$*, and thus the terminology* partial Gaussian mixture *is sometimes employed.)*

### D.1  Constructing an Outer Bracket

The first step is to show that pushing a mean far away from a region will rapidly decrease its density there, which is immediate from the condition $\sigma_1 I \preceq \Sigma \preceq \sigma_2 I$.

**Lemma D.2.** *Let probability measure* $\rho$*, accuracy* $\epsilon > 0$*, covariance lower bound* $0 < \sigma_1 \leq \sigma_2$*, and radius* $R$ *with corresponding* $l_2$ *ball* $B := \{x \in \mathbb{R}^d : \|x - \mathbb{E}_\rho(X)\|_2 \leq R\}$ *be given. Define*

$$R_1 := \sqrt{2\sigma_2 \ln \left( \frac{1}{(2\pi\sigma_1)^{d/2} \epsilon^2} \right)}$$

$$R_2 := R + R_1,$$

$$B_2 := \{\mu \in \mathbb{R}^d : \|\mu - \mathbb{E}_\rho(X)\|_2 \leq R_2\}.$$

*If* $\theta = (\mu, \Sigma)$ *is the parameterization of a Gaussian density* $p_\theta$ *with* $\sigma_1 I \preceq \Sigma \preceq \sigma_2 I$ *but* $\mu \notin B_2$*, then* $p_\theta(x) < \epsilon$ *for every* $x \in B$.

*Proof.* Let Gaussian parameters $\theta = (\mu, \Sigma)$ be given with $\sigma_1 I \preceq \Sigma \preceq \sigma_2 I$, but $\mu \notin B_2$. By the definition of $B_2$, for any $x \in B_1$,

$$p_\theta(x) < (2\pi\sigma_1)^{-d/2} \exp(-R_1^2/(2\sigma_2)) = \epsilon.$$

$\square$

The upper component of the outer bracket will be constructed first (and indeed used in the construction of the lower component).

**Lemma D.3.** *Let probability measure $\rho$ with order-$p$ moment bound with respect to $\|\cdot\|_2$, target accuracy $\epsilon > 0$, and covariance lower bound $0 < \sigma_1$ be given. Define*

$$p_{\max} := (2\pi\sigma_1)^{-d/2},$$
$$u(x) := \ln(p_{\max}),$$
$$R_u := (M|\ln(p_{\max})|/\epsilon)^{1/p},$$
$$B_u := \left\{ x \in \mathbb{R}^d : \|x - \mathbb{E}_\rho(X)\|_2 \le R_u \right\}.$$

*If $p_\theta$ denotes a Gaussian density with parameters $\theta = (\mu, \Sigma)$ satisfying $\Sigma \succeq \sigma_1 I$, then $p_\theta \le u$ everywhere. Additionally,*

$$\left| \int_{B_u^c} u(x) d\rho(x) \right| \le \int_{B_u^c} |u(x)| d\rho(x) \le \epsilon,$$

*and with probability at least $1 - \delta$ over the draw of $m$ points from $\rho$,*

$$\left| \int_{B_u^c} u(x) d\hat{\rho}(x) \right| \le \int_{B_u^c} |u(x)| d\hat{\rho}(x) \le \epsilon + |\ln(p_{\max})|\sqrt{\frac{1}{2m}\ln\left(\frac{1}{\delta}\right)}.$$

*(That is to say, $u$ is the upper part of an outer bracket for all Gaussians (and mixtures thereof) where each covariance $\Sigma$ satisfies $\Sigma \succeq \sigma_1 I$.)*

*Proof.* Let $p_\theta$ with $\theta = (\mu, \Sigma)$ satisfying $\Sigma \succeq \sigma_1 I$ be given. Then

$$p_\theta(x) \le \frac{1}{\sqrt{(2\pi)^d \sigma_1^d}} \exp(\,0\,) = p_{\max}.$$

Next, given the form of $B_u$, if $\ln(p_{\max}) = 0$, the result is immediate, thus suppose $\ln(p_{\max}) \ne 0$; Lemma A.2 provides that $\rho(B_u) \ge 1 - \epsilon/|\ln(p_{\max})|$, whereby

$$\left| \int_{B_u^c} u(x) d\rho(x) \right| \le \int_{B_u^c} |u(x)| d\rho(x) = |\ln(p_{\max})|\rho(B_u^c) \le \epsilon.$$

For the finite sample guarantee, by Hoeffding's inequality,

$$\hat{\rho}(B_u^c) \le \rho(B_u^c) + \sqrt{\frac{1}{2m}\ln\left(\frac{1}{\delta}\right)} \le \frac{\epsilon}{|\ln(p_{\max})|} + \sqrt{\frac{1}{2m}\ln\left(\frac{1}{\delta}\right)},$$

which gives the result similarly to the case for $\rho$. $\qquad\square$

From, here, a tiny control on $\mathcal{S}_{\mathrm{mog}}(\nu; c, k, \sigma_1, \sigma_2)$ emerges, analogous to Lemma B.2 for $\mathcal{H}_f(\nu; c, k)$.

**Lemma D.4.** *Let covariance bounds $0 < \sigma_1 \le \sigma_2$, cost $c \le 1/2$, and probability measure $\rho$ with order-$p$ moment bound $M$ with respect to $\|\cdot\|_2$ be given. Define*

$$p_{\max} := (2\pi\sigma_1)^{-d/2},$$
$$R_3 := (2M|\ln(p_{\max})|)^{1/p},$$
$$R_4 := (2M)^{1/p},$$
$$R_5 := \sqrt{2\sigma_2\left(\ln\left(\frac{8e}{(2\pi\sigma_1)^{d/2}}\right) - 4c\right)}.$$
$$R_6 := \max\{R_3, R_4\} + R_5.$$
$$B_6 := \{x \in \mathbb{R}^d : \|x - \mathbb{E}_\rho(X)\|_2 \le R_6\}.$$

*Suppose*

$$m \ge 2\ln(1/\delta)\max\{4, |\ln(p_{\max})|^2\}.$$

*With probability at least $1 - 2\delta$, given any $(\alpha, \Theta) \in \mathcal{S}_{\mathrm{mog}}(\rho; c, k, \sigma_1, \sigma_2) \cup \mathcal{S}_{\mathrm{mog}}(\hat{\rho}; c, k, \sigma_1, \sigma_2)$, the restriction $(\alpha', \Theta') = (\alpha, \Theta) \sqcap B_6$ is nonempty, and moreover satisfies $\sum_{\alpha_i \in \alpha'} \alpha_i \ge \exp(4c)/(8ep_{\max})$.*

*Proof.* Define

$$B_3 := \left\{ x \in \mathbb{R}^d : \|x - \mathbb{E}_\rho(X)\|_2 \le \max\{R_3, R_4\} \right\}.$$

Since $B_3$ has radius at least $R_4$, Lemma A.2 provides

$$\rho(B_3) \ge 1/2,$$

and Hoeffding's inequality and the lower bound on $m$ provide (with probability at least $1 - \delta$)

$$\hat{\rho}(B_3) \ge \frac{1}{2} - \sqrt{\frac{2}{m} \ln\left(\frac{1}{\delta}\right)} > \frac{1}{4}.$$

Additionally, since $B_3$ also has radius at least $R_3$, by Lemma D.3, the choice of $B_3$, and the lower bound on $m$, and letting $B_4$ denote the ball of radius $R_3$,

$$\left| \int_{B_3^c} u\,d\rho \right| \le \int_{B_4^c} |u|\,d\rho \le \int_{B_4^c} |u|\,d\rho \le 1/2 \qquad \text{and} \qquad \left| \int_{B_3^c} u\,d\hat{\rho} \right| < 1,$$

where the statement for $\hat{\rho}$ is with probability at least $1 - \delta$. For the remainder of the proof, let $\nu$ refer to either $\rho$ or $\hat{\rho}$, and discard the $2\delta$ failure probability of either of the above two events.

For convenience, define $p_0 := \exp(4c)/(8e)$, whereby

$$R_5 = \sqrt{2\sigma_2 \ln\left(\frac{1}{p_0(2\pi\sigma_1)^{d/2}}\right)}.$$

By Lemma D.2, any Gaussian parameters $\theta = (\mu, \Sigma)$ with $\sigma_1 I \preceq \Sigma \preceq \sigma_2 I$ and $\mu \notin B_6$ have $p_\theta(x) < p_0$ everywhere on $B_3$. As such, a mixture $(\alpha, \Theta)$ where each $\theta_i \in \Theta$ satisfies these conditions also satisfies

$$\int \ln\left(\sum_i \alpha_i p_{\theta_i}\right) d\nu \le \int_{B_3} \ln\left(\sum_{(\alpha_i,\theta_i) \in (\alpha,\Theta) \sqcap B_6} \alpha_i p_{\theta_i} + \sum_{(\alpha_i,\theta_i) \notin (\alpha,\Theta) \sqcap B_6} \alpha_i p_{\theta_i}\right) d\nu + \int_{B_3^c} u\,d\nu$$

$$< \ln\left(\sum_{(\alpha_i,\theta_i) \in (\alpha,\Theta) \sqcap B_6} \alpha_i p_{\max} + \sum_{\alpha_i,\theta_i) \notin (\alpha,\Theta) \sqcap B_6} \alpha_i p_0\right) \nu(B_3) + 1$$

Suppose contradictorily that $(\alpha, \Theta) \sqcap B_6 = \emptyset$ or $\sum_{(\alpha_i,\theta_i) \in (\alpha,\Theta) \sqcap B_6} \alpha_i < p_0/p_{\max}$. But $c \le 1/2$ implies $p_0 \le 1/2$ and so $\ln(2p_0) \le 0$, thus $\ln(2p_0)\nu(B_3) \le \ln(2p_0)/4$ which together with $p_0 \le \exp(4c)/(8e)$ and the above display gives

$$\int \ln\left(\sum_i \alpha_i p_{\theta_i}\right) d\nu < \ln(2p_0)/4 + 1 \le c,$$

which contradicts $\mathbb{E}_\nu(\phi_{\mathrm{g}}(X; (\alpha, \Theta))) \ge c$. $\qquad\qquad \square$

Now that significant weight can be shown to reside in some restricted region, the outer bracket and its basic properties follow (i.e., the analog to Lemma 3.5).

**Lemma D.5.** *Let target accuracy $0 < \epsilon \le 1$, covariance bounds $0 < \sigma_1 \le \sigma_2$ with $\sigma_1 \le 1$, target cost $c$, confidence parameter $\delta \in (0, 1]$, probability measure $\rho$ with order-p moment bound $M$ with respect to $\|\cdot\|_2$ with $p \ge 4$, and integer $1 \le p' \le p/2 - 1$. Define first the basic quantities*

$$M' := 2^{p'}\epsilon,$$

$$p_{\max} := (2\pi\sigma_1)^{-d/2},$$

$$R_6 := (2M|\ln(p_{\max})|)^{1/p} + (2M)^{1/p} + \sqrt{2\sigma_2\left(\ln\left(\frac{8e}{(2\pi\sigma_1)^{d/2}}\right) - 4c\right)},$$

$$B_6 := \{x \in \mathbb{R}^d : \|x - \mathbb{E}_\rho(X)\|_2 \le R_6\|\}.$$

*Additionally define the outer bracket elements*

$$Z_\ell := \left\{ (\alpha, \Theta) : \forall (\alpha_i, (\mu_i, \theta_i)) \in (\alpha, \Theta) \centerdot \mu_i \in B_6, \sigma_1 I \preceq \Sigma \preceq \sigma_2 I, \sum_i \alpha_i \geq \exp(4c)/(8ep_{\max}) \right\},$$

$$c_\ell := 4c - \ln(8ep_{\max}) - \frac{d}{2}\ln(2\pi\sigma_2),$$

$$\ell(x) := c_\ell - \frac{2}{\sigma_1} \|x - \mathbb{E}_\rho(X)\|_2^2,$$

$$u(x) := \ln(p_{\max}),$$

$$\epsilon_{\hat{\rho}} := \epsilon + (|c_\ell| + |\ln(p_{\max})|)\sqrt{\frac{1}{2m}\ln\left(\frac{1}{\delta}\right)} + \sqrt{\frac{M'ep'}{2m}}\left(\frac{2}{\delta}\right)^{1/p'},$$

$$M_1 := (2M|c_\ell|)^{1/p} + (4M\sigma_1)^{1/(p-2)} + \max_{1 \leq i \leq p'} M^{1/(p-2i)} + (M|\ln(p_{\max})|)^{1/p},$$

$$R_B = R_6 + M_1/\epsilon^{1/(p-2p')},$$

$$B := \{x \in \mathbb{R}^d : \|x - \mathbb{E}_\rho(X)\|_2 \leq R_B\}.$$

*The following statements hold with probability at least $1 - 4\delta$ over a draw of size*

$$m \geq \max\left\{p'/(M'e), 8\ln(1/\delta), 2|\ln(p_{\max})|^2\ln(1/\delta)\right\}.$$

1. *$(u, \ell)$ is an outer bracket for $\rho$ at scale $\epsilon_\rho := \epsilon$ with sets $B_\ell := B_u := B$, center set class $Z_\ell$ as above, and $Z_u = \mathcal{S}_{\text{mog}}(\rho; \infty, k, \sigma_1, \sigma_2)$. Additionally, $(u, \ell)$ is also an outer bracket for $\hat{\rho}$ at scale $\epsilon_{\hat{\rho}}$ with the same sets.*

2. *Define*

$$R_C := 1 + R_B(1 + \sqrt{8\sigma_2/\sigma_1}) + \sqrt{4\sigma_2 \ln(1/\epsilon)} + \sqrt{2\sigma_2\left(\ln\left(\frac{64e^2(2\pi\sigma_2)^d}{(2\pi)^d p_{\max}^4}\right) - 8c\right)},$$

$$C := \{\mu \in \mathbb{R}^d : \|x - \mathbb{E}_\rho(X)\|_2 \leq R_C\}.$$

*Every $(\alpha, \Theta) \in \mathcal{S}_{\text{mog}}(\rho; c, k, \sigma_1, \sigma_2) \cup \mathcal{S}_{\text{mog}}(\hat{\rho}; c, k, \sigma_1, \sigma_2)$ satisfies $\sum_{(\alpha_i, \theta_i) \in (\alpha, \Theta) \sqcap C} \alpha_i \geq \exp(4c)/(8ep_{\max})$, and*

$$\left| \int \phi_g(x; (\alpha, \Theta)) d\rho(x) - \int_B \phi_g(x; (\alpha, \Theta) \sqcap C) d\rho(x) \right| \leq \epsilon_\rho = 2\epsilon$$

*and*

$$\left| \int \phi_g(x; (\alpha, \Theta)) d\hat{\rho}(x) - \int_B \phi_g(x; (\alpha, \Theta) \sqcap C) d\hat{\rho}(x) \right| \leq \epsilon + \epsilon_{\hat{\rho}}.$$

*Proof of Lemma D.5.* It is useful to first expand the choice of $R_B$, which was chosen large enough to carry a collection of other radii. In particular, since $\epsilon \leq 1$, then $1/\epsilon \geq 1$, and therefore $1/\epsilon^a \leq 1/\epsilon^b$ when $a \leq b$. As such, since $p' \leq p/2 - 1$,

$$R_B = R_6 + M_1/\epsilon^{1/(p-2p')}$$

$$= R_6 + \left((2M|c_\ell|)^{1/p} + (4M\sigma_1)^{1/(p-2)} + \max_{1 \leq i \leq p'} M^{1/(p-2i)} + (M|\ln(p_{\max})|)^{1/p}\right)/\epsilon^{1/(p-2p')}$$

$$\geq R_6 + \left((2M|c_\ell|/\epsilon)^{1/p} + (4M\sigma_1/\epsilon)^{1/(p-2)} + \max_{1 \leq i \leq p'}(M/\epsilon)^{1/(p-2i)} + (M|\ln(p_{\max})|/\epsilon)^{1/p}\right).$$

Since every term is nonnegative, $R_B$ dominates each individual term.

1. The upper bracket and its guarantees were provided by Lemma D.3; note that $\epsilon_{\hat{\rho}}$ is defined large enough to include the deviations there, and similarly $R_B \geq (M|\ln(p_{\max})|/\epsilon)^{1/p}$ means the $B$ here is defined large enough to contain the $B_u$ there; correspondingly, discard a failure event with probability mass at most $\delta$.

Let the lower bracket be defined as in the statement; note that its properties are much more conservative as compared with the upper bracket. Let $(\alpha, \Theta) \in Z_\ell$ be given. For every $\theta_i = (\mu_i, \Sigma_i)$, $\|\mu_i - \mathbb{E}_\rho(X)\|_2 \leq R_6$, whereas $R_B \geq R_6$ meaning $x \in B^c$ implies $\|x - \mathbb{E}_\rho(X)\|_2 \geq R_6$, so

$$\|x - \mu_i\|_2 \leq \|x - \mathbb{E}_\rho(X)\|_2 + \|\mu_i - \mathbb{E}_\rho(X)\|_2 \leq 2\|x - \mathbb{E}_\rho(X)\|_2,$$

which combined with $\sigma_1 I \preceq \Sigma_i \preceq \sigma_2 I$ gives

$$\ln\left(\sum_i \alpha_i p_{\theta_i}(x)\right) \geq \ln\left(\sum_i \alpha_i \frac{1}{(2\pi\sigma_2)^{d/2}} \exp\left(-\frac{1}{2\sigma_1}\|x - \mu_i\|_2^2\right)\right)$$

$$\geq \ln(p_0/p_{\max}) - \frac{d}{2}\ln(2\pi\sigma_2) - \frac{2}{\sigma_1}\|x - \mathbb{E}_\rho(X)\|_2^2$$

$$= \ell(x),$$

which is the dominance property.

Next come the integral properties of $\ell$. By Lemma A.2 and since $R_B \geq (2M|c_\ell|/\epsilon)^{1/p}$,

$$\left|\int_{B^c} c_\ell d\rho\right| \leq \int_{B^c} |c_\ell| d\rho \leq \int_{B^c} |c_\ell| d\rho = \rho(B^c)|c_\ell| \leq \epsilon/2.$$

Similarly, by Hoeffding's inequality, with probability at least $1 - \delta$,

$$\left|\int_{B_\ell^c} c_\ell d\hat\rho\right| \leq \epsilon/2 + |c_\ell|\sqrt{\frac{1}{2m}\ln\left(\frac{1}{\delta}\right)}.$$

Now define

$$\ell_1(x) := -\frac{2}{\sigma_1}\|x - \mathbb{E}_\rho(X)\|_2^2 = \ell(x) - c_\ell.$$

By Lemma A.6 and since $R_B \geq (4\sigma_1 M/\epsilon)^{1/(p-2)}$,

$$\left|\int_{B^c} \ell_1 d\rho\right| \leq \int_{B^c} |\ell_1| d\rho = \frac{2}{\sigma_1}\int_{B^c} \|x - \mathbb{E}_\rho(X)\|_2^2 d\rho(x) \leq \epsilon/2.$$

Furthermore by Lemma A.7 and the above estimate, and since $R_B \geq \max_{1 \leq i \leq p'}(M/\epsilon)^{1/(p-2i)}$ (where the maximum is attained at one of the endpoints), then with probability at least $1 - \delta$

$$\left|\int_{B^c} \ell_1 d\hat\rho\right| \leq \frac{\epsilon}{2} + \sqrt{\frac{M'ep'}{2m}}\left(\frac{2}{\delta}\right)^{1/p'}.$$

Unioning together the above failure probabilities, the general controls for $\ell = c_\ell + \ell_1$ follow by the triangle inequality and definition of $\epsilon_{\hat\rho}$.

2. Throughout the following, let $\nu$ denote either $\rho$ or $\hat\rho$, and correspondingly let $\epsilon_\nu$ respectively refer to $\epsilon_\rho$ or $\epsilon_{\hat\rho}$; let the above bracketing properties hold throughout (with events appropriately discarded for $\hat\rho$). Furthermore, for convenience, define

$$p_0 := \exp(4c)/(8e).$$

Let any $(\alpha, \Theta)$ be given with $(\alpha, \Theta) \in \mathcal{S}_{\mathrm{mog}}(\rho; c, k, \sigma_1, \sigma_2) \cup \mathcal{S}_{\mathrm{mog}}(\hat\rho; c, k, \sigma_1, \sigma_2)$. Define the two index sets

$$I_C := \{i \in [k] : (\alpha_i, \theta_i) \in (\alpha, \Theta) \sqcap C\},$$
$$I_6 := \{i \in [k] : (\alpha_i, \theta_i) \in (\alpha, \Theta) \sqcap B_6\}.$$

By Lemma D.4, with probability at least $1 - \delta$, $\sum_{i \in I_6} \alpha_i \geq p_0/p_{\max}$; henceforth discard the corresponding failure event, bringing the total discarded probability mass to $4\delta$.

To start, since $\ln(\cdot)$ is concave and thus $\ln(a + b) \leq \ln(a) + b/a$ for any positive $a, b$,

$$\int \ln \left( \sum_i \alpha_i p_{\theta_i}(x) \right) d\nu(x) \leq \int_B \ln \left( \sum_i \alpha_i p_{\theta_i}(x) \right) d\nu(x) + \int_{B^c} u(x) d\nu(x)$$

$$\leq \int_B \ln \left( \sum_{i \in I_C} \alpha_i p_{\theta_i}(x) \right) d\nu(x) + \int_B \frac{\sum_{i \notin I_C} \alpha_i p_{\theta_i}(x)}{\sum_{i \in I_C} \alpha_i p_{\theta_i}(x)} d\nu(x) + \epsilon_\nu.$$

In order to control the fraction, both the numerator and denominator will be uniformly controlled for every $x \in B$, whereby the result follows since $\nu$ is a probability measure (i.e., the integral is upper bounded with an upper bound on the numerator times $\nu(B) \leq 1$ divided by a lower bound on the denominator).

For the purposes of controlling this fraction, define

$$p_1 := \frac{1}{(2\pi\sigma_2)^{d/2}} \exp \left( -\frac{R_B^2 + R_6^2}{\sigma} \right),$$

$$p_2 := \epsilon p_1 p_0 / p_{\max},$$

Observe, by choice of $R_C$ and since $\sigma_1 \leq 1$, that

$$R_B + \sqrt{2\sigma_2 \ln \left( \frac{1}{p_2^2 (2\pi)^d \sigma_1^{d-1}} \right)} \leq R_B + \sqrt{2\sigma_2 \ln \left( \frac{64 e^2 p_{\max}^2 (2\pi\sigma_2)^d \exp(2(R_B^2 + R_6^2))}{\epsilon^2 \exp(8c)(2\pi)^d \sigma_1^d} \right)}$$

$$\leq R_B + \sqrt{2\sigma_2 \left( \ln \left( \frac{64 e^2 (2\pi\sigma_2)^d}{\epsilon^2 (2\pi)^d p_{\max}^4} \right) - 8c - 4R_B^2/\sigma \right)}$$

$$\leq R_B + \sqrt{2\sigma_2 \left( \ln \left( \frac{64 e^2 (2\pi\sigma_2)^d}{(2\pi)^d p_{\max}^4} \right) - 8c \right)}$$

$$+ \sqrt{4\sigma_2 \ln(1/\epsilon)} + R_B \sqrt{8\sigma_2/\sigma_1}$$

$$\leq R_C.$$

For the denominator, first note for every $x \in B$ and parameters $\theta = (\mu, \Sigma)$ with $\sigma_1 I \preceq \Sigma \preceq \sigma_2 I$ and $\mu \in B_6$ that

$$p_\theta(x) \geq \frac{1}{(2\pi\sigma_2)^{d/2}} \exp \left( -\frac{1}{2\sigma_1} \|x - \mu\|_2^2 \right)$$

$$\geq \frac{1}{(2\pi\sigma_2)^{d/2}} \exp \left( -\frac{1}{2\sigma_1} \left( \|x - \mathbb{E}_\rho(X)\|_2 + \|\mathbb{E}_\rho(X) - \mu_i\|_2 \right)^2 \right)$$

$$\geq p_1.$$

Consequently, for $x \in B$,

$$\sum_{i \in I_C} \alpha_i p_i(x) \geq \sum_{i \in I_6} \alpha_i p_i(x) \geq p_1 \sum_{i \in I_6} \alpha_i \geq p_1 p_0 / p_{\max}.$$

For the numerator, by choice of $C$ (as developed above with the definitions of $p_1$ and $p_2$) and an application of Lemma D.2, for $p_i$ corresponding to $i \notin I_C$,

$$p_i(x) \leq \epsilon p_1 p_0 / p_{\max} = p_2.$$

It follows that the fractional term is at most $\epsilon$, which gives the first direction of the desired inequality.

To get the other direction, since $\sum_{i \in I_6} \alpha_i \geq p_0 / p_{\max}$ due to Lemma D.4 as discussed above, it follows that $(\alpha, \Theta) \sqcap B_6 \in Z_\ell$, meaning the corresponding partial Gaussian mixture can be controlled by $\ell$. As such, since $R_6 \leq R_B$ thus $I_6 \subseteq I_C$, and since $\ln$ is

nondecreasing,

$$\int_B \ln\left(\sum_{i\in I_C}\alpha_i p_i\right) d\nu = \int \ln\left(\sum_{i\in I_C}\alpha_i p_i\right) d\nu - \int_{B^c}\ln\left(\sum_{i\in I_C}\alpha_i p_i\right) d\nu$$

$$\leq \int \ln\left(\sum_{i\in I_C}\alpha_i p_i\right) d\nu - \int_{B^c}\ln\left(\sum_{i\in I_6}\alpha_i p_i\right) d\nu$$

$$\leq \int \ln\left(\sum_{i\in I_C}\alpha_i p_i\right) d\nu - \int_{B^c}\ell\, d\nu$$

$$\leq \int \ln\left(\sum_{i\in I_C}\alpha_i p_i\right) d\nu + \epsilon_\nu$$

$$\leq \int \ln\left(\sum_i \alpha_i p_i\right) d\nu + \epsilon_\nu.$$

$\square$

## D.2 Uniform Covering of Gaussian Mixtures

First, a helper lemma for covering covariance matrices.

**Lemma D.6.** *Let scale $\epsilon > 0$ and eigenvalue bounds $0 < \sigma_1 \leq \sigma_2$ be given. There exists a subset $\mathcal{M}$ of the positive definite matrices satisfying $\sigma_1 I \preceq M \preceq \sigma_2 I$ so that*

$$|\mathcal{M}| \leq (1 + 32\sigma_2/\epsilon)^{d^2}\left(\left(1 + \frac{\sigma_2 - \sigma_1}{\epsilon/2}\right)^d + \left(\frac{\ln(\sigma_2/\sigma_1)}{\epsilon/d}\right)^d\right),$$

*and for any $A$ with $\sigma_1 I \preceq A \preceq \sigma_2 I$, there exists $B \in \mathcal{M}$ with*

$$\exp(-\epsilon) \leq \frac{|A|}{|B|} \leq \exp(\epsilon) \qquad \text{and} \qquad \|A - B\|_2 \leq \epsilon.$$

*Proof.* The mechanism of the proof is to separately cover the set of orthogonal matrices and the set of possible eigenvalues; this directly leads to the determinant control, and after some algebra, the spectral norm control follows as well.

With foresight, set the scales

$$\tau := \epsilon/(8\sigma_2),$$
$$\tau' := \epsilon/2,$$
$$\tau'' := \exp(\epsilon/d).$$

First, a cover of the orthogonal $d \times d$ matrices at scale $\tau$ is constructed as follows. The entries of these orthogonal matrices are within $[-1, +1]$, thus first construct a cover $Q'$ of all matrices $[-1, +1]^{d \times d}$ at scale $\tau/2$ according to the maximum-norm, which simply measures the max among entrywise differences; this cover can be constructed by gridding each coordinate at scale $\tau/2$, and thus

$$|Q'| \leq (1 + 4/\tau)^{d^2}.$$

Now, to produce a cover of the orthogonal matrices, for each $M' \in Q'$, if it is within max-norm distance $\tau/2$ of some orthogonal matrix $M$, include $M$ in the new cover $Q$; otherwise, ignore $M'$. Since $Q'$ was a max-norm cover of $[-1, +1]^{d \times d}$ at scale $\tau/2$, then $Q$ must be a max-norm cover of the orthogonal matrices at scale $\tau$ (by the triangle inequality), and it still holds that

$$|Q| \leq (1 + 4/\tau)^{d^2}.$$

Since the max-norm is dominated by the spectral norm, for any orthogonal matrix $O$, there exists $Q \in M$ with $\|O - Q\|_2 \leq \tau$.

Second, a cover of the set of possible eigenvalues is constructed as follows; since both a multiplicative and an additive guarantee are needed for the eigenvalues, two covers will be unioned together. First, produce a cover $L_1$ of the set $[\sigma_1, \sigma_2]^d$ at scale $\tau'$ entrywise as usual, which means $|L_1| \leq (1 + (\sigma_2 - \sigma_1)/\tau')^d$. Second, the cover $L_2$ will cover each coordinate multiplicatively, meaning each coordinate cover consists of $\sigma_1, \sigma_1\tau'', \sigma_1(\tau'')^2$, and so on; consequently, this cover has size $|L_2| \leq \ln(\sigma_2/\sigma_1)/\ln(\tau'')$. Together, the cover $L := L_1 \cup L_2$ has size

$$|L| \leq \left(1 + \frac{\sigma_2 - \sigma_1}{\tau'}\right)^d + \left(\frac{\ln(\sigma_2/\sigma_1)}{\ln(\tau'')}\right)^d,$$

and for any vector $\Lambda \in [\sigma_1, \sigma_2]^d$, there exists $\Lambda' \in L$ with

$$\frac{1}{\tau''} \leq \max_i \Lambda'_i/\Lambda_i \leq \tau'' \qquad \text{and} \qquad \max_i |\Lambda'_i - \Lambda_i| \leq \tau.$$

Note there was redundancy in this construction: $L$ need only contain nondecreasing sequences.

The final cover $\mathcal{M}$ is thus the cross product of $Q$ and $L$, and correspondingly its size is the product of their sizes. Given any $A$ with $\sigma_1 I \preceq A \preceq \sigma_2 I$ with spectral decomposition $O_1^\top \Lambda_1 O_1$, pick a corresponding $O_2 \in Q$ which is closest to $O_1$ in spectral norm, and $\Lambda_2 \in L$ which is closest to $\Lambda_1$ in max-norm, and set $B = O_2^\top \Lambda_2 O_2$. By the multiplicative guarantee on $L$, it follows that

$$\left(\frac{1}{\tau''}\right)^d \leq \frac{|\Lambda_2|}{|\Lambda_1|} = \frac{|B|}{|A|} \leq (\tau'')^d;$$

by the choice of $\tau''$, the determinant guarantee follows. Secondly, relying on a few properties of spectral norms ($\|XY\|_2 \leq \|X\|_2\|Y\|_2$ for square matrices, and $\|Z\|_2 = 1$ for orthogonal matrices, and of course the triangle inequality),

$$\|A - B\|_2 = \left\|(O_1 - O_2 + O_2)^\top \Lambda_1 (O_1 - O_2 + O_2)^\top - O_2^\top \Lambda_2 O_2\right\|_2$$
$$\leq \|O_2^\top \Lambda_1 O_2 - O_2^\top \Lambda_2 O_2\|_2 + 2\|O_2^\top \Lambda_1 (O_1 - O_2)\|_2 + \|(O_1 - O_2)^\top \Lambda_1 (O_1 - O_2)\|_2$$
$$\leq \|\Lambda_1 - \Lambda_2\|_2 + 2\|O_1 - O_2\|_2\|\Lambda_1\|_2 + \|O_1 - O_2\|_2\|\Lambda_1\|_2(\|O_1\|_2 + \|O_2\|_2)$$
$$\leq \tau' + 4\tau\sigma_2,$$

and the second guarantee follows by choice of $\tau$ and $\tau'$. $\qquad\square$

The covering lemma is as follows.

**Lemma D.7.** *Let scale $\epsilon > 0$, ball $B := \{x \in \mathbb{R}^d : \|x - \mathbb{E}(X)\| \leq R\}$, mean set $X := \{x \in \mathbb{R}^d : \|x - \mathbb{E}(X)\| \leq R_2\}$, covariance eigenvalue bounds $0 < \sigma_1 \leq \sigma_2$, mass lower bound $c_1 > 0$, and number of mixtures $k > 0$ be given. Then there exists a cover set $\mathcal{N}$ (where $(\mu, \Sigma) \in \mathcal{N}$ has $\mu \in X$ and $\sigma_1 I \preceq \Sigma \preceq \sigma_2 I$) of size*

$$|\mathcal{N}| \leq \left(\left(\frac{\ln(1/\alpha_0)}{\ln(\tau_0)} + \frac{1 - \alpha_0}{\tau_4}\right) \cdot \left(1 + \frac{2R_2 d}{\tau_1}\right)^d \cdot (1 + 32/(\sigma_1\tau_2))^{d^2}\left(\left(1 + \frac{\sigma_1^{-1} - \sigma_2^{-1}}{\tau_2/2}\right)^d + \left(\frac{\ln(\sigma_2/\sigma_1)}{\tau_2/d}\right)^d\right)\right)^k$$

*where*

$$\tau_0 := \exp(\epsilon/4),$$
$$\tau_1 := \min\left\{\frac{\epsilon\sigma_1}{16(R + R_2)}, \sqrt{\frac{\epsilon\sigma_1}{8}}\right\},$$
$$\tau_2 := \frac{\epsilon}{4\max\{1, (R + R_2)^2\}},$$
$$p_{\min} := \frac{1}{(2\pi\sigma_2)^{d/2}}\exp(-(R + R_2)^2/(2\sigma_1)),$$
$$p_{\max} := (2\pi\sigma_1)^{-d/2},$$
$$\alpha_0 := \frac{\epsilon c_1 p_{\min}}{4k(p_{\max} + \epsilon p_{\min}/2)},$$
$$\tau_4 := \alpha_0,$$

*(whereby $p_{\min} \leq p_\theta(x) \leq p_{\max}$ for $x \in B$ and $\theta = (\mu, \Sigma)$ satisfies $\mu \in X$ and $\sigma_1 I \preceq \Sigma \preceq \sigma_2 I$,) so that for every partial Gaussian mixture $(\alpha, \Theta) = \{(\alpha_i, \mu_i, \Sigma_i)\}$ with $\alpha_i \geq 0$, $c_1 \leq \sum_i \alpha_i \leq 1$, $\mu_i \in X$, and $\sigma_1 I \preceq \Sigma_i \preceq \sigma_2 I$ there is an element $(\alpha', \Theta') \in \mathcal{N}$ with weights $c_1 - k\alpha_0 \leq \sum_i \alpha'_i \leq 1$ so that, for every $x \in B$,*

$$|\ln(p_{\alpha,\Theta}(x)) - \ln(p_{\alpha',\Theta'}(x))| \leq \epsilon.$$

*Proof.* The proof controls components in two different ways. For those where the weight $\alpha_i$ is not too small, both $\alpha_i$ and $p_{\theta_i}$ are closely (multiplicatively) approximated. When $\alpha_i$ is small, its contribution can be discarded. Between these two cases, the bound follows.

Note briefly that for any $\theta = (\mu, \Sigma)$ with $\mu \in X$ and $\sigma_1 I \preceq \Sigma \preceq \sigma_2 I$,

$$p_\theta(x) \leq \frac{1}{(2\pi\sigma_1)^{d/2}} \exp(\, 0 \,) = p_{\max},$$

$$p_\theta(x) \geq \frac{1}{(2\pi\sigma_2)^{d/2}} \exp(-\|x - \mu\|_2^2/(2\sigma_1))$$

$$\geq \frac{1}{(2\pi\sigma_2)^{d/2}} \exp(-(\|x - \mathbb{E}_\rho(X)\|_2 + \|\mu - \mathbb{E}_\rho(X)\|)^2/(2\sigma_1))$$

$$= p_{\min}.$$

Next, the covers of each element of the Gaussian mixture are as follows.

1. Union together a multiplicatively grid of $[\alpha_0, 1]$ at scale $\tau_0$ (meaning produce a sequence of the form $\alpha_0, \alpha_0\tau_0, \alpha_0\tau_0^2$, and so on), and an additive grid of $[\alpha_0, 1]$ at scale $\tau_4$; together, the grid has a size of at most
$$\frac{\ln(1/\alpha_0)}{\ln(\tau_0)} + \frac{1 - \alpha_0}{\tau_4}.$$

2. Grid the candidate center set $X$ at scale $\tau_1$, which by Lemma B.5 can be done with size at most
$$\left(1 + \frac{2R_2 d}{\tau_1}\right)^d.$$

3. Lastly, grid the *inverse* of covariance matrices (sometimes called precision matrices), meaning $\sigma_2^{-1} I \preceq \Sigma^{-1} \preceq \sigma_1^{-1}$, whereby Lemma D.6 grants that a cover of size
$$(1 + 32/(\sigma_1\tau_2))^{d^2} \left(\left(\frac{\sigma_1^{-1} - \sigma_2^{-1}}{\tau_2/2}\right)^d + \left(\frac{\ln(\sigma_1/\sigma_2)}{\tau_2/d}\right)^d\right)$$
suffices to provide that for any permissible $\Sigma^{-1}$, there exists a cover element $A$ with
$$\exp(-\tau_2) \leq \frac{|\Sigma^{-1}|}{|A|} \leq \exp(\tau_2) \qquad \text{and} \qquad \|\Sigma^{-1} - A\|_2 \leq \tau_2.$$

Producing the size of these various covers and raising to the power $k$ (to handle at most $k$ components), the cover size in the statement is met.

Now consider a component $(\alpha_i, \mu_i, \Sigma_i)$ with $\alpha_i \geq \alpha_0$; a relevant cover element $(a_i, c_i, B_i)$ is chosen as follows.

1. Choose the largest $a_i \leq \alpha_i$ in the gridding of $[\alpha_0, 1]$, whereby it follows that $\sum_i a_i \leq \sum_i \alpha_i \leq 1$, and also
$$\tau_0^{-1} \leq a_i/\alpha_i \leq \tau_0 \qquad \text{and} \qquad a_i \geq \alpha_i - \tau_4.$$
Thanks to the second property,
$$\sum_{\alpha_i \geq \alpha_0} a_i \geq \left(\sum_{\alpha_i \geq \alpha_0} \alpha_i\right) - k\tau_4.$$

2. Choose $c_i$ in the grid on $X$ so that $\|\mu_i - c_i\| \leq \tau_1$.

3. Choose covariance $B_i$ so that

$$\exp(-\tau_2) \leq \frac{|B_i|}{|\Sigma_i|} \leq \exp(\tau_2) \qquad \text{and} \qquad \|\Sigma^{-1} - B_i^{-1}\|_2 \leq \tau_2.$$

The first property directly controls for the determinant term in the Gaussian density. To control the Mahalanobis term, note that the above display, combined with $\|\mu_i - c_i\| \leq \tau_1$, gives, for every $x \in B$,

$$
\begin{aligned}
&\left| (x - \mu_i)^\top \Sigma_i^{-1} (x - \mu_i) - (x - c_i)^\top B_i^{-1} (x - c_i) \right| \\
&= \left| (x - \mu_i)^\top \Sigma_i^{-1} (x - \mu_i) - (x - c_i)^\top (B_i^{-1} - \Sigma_i^{-1} + \Sigma_i^{-1})(x - c_i) \right| \\
&\leq \left| (x - \mu_i)^\top \Sigma_i^{-1} (x - \mu_i) - (x - c_i)^\top \Sigma_i^{-1} (x - c_i) \right| + \|x - c_i\|_2^2 \|B_i^{-1} - \Sigma_i^{-1}\|_2 \\
&\leq \left| (x - \mu_i)^\top \Sigma_i^{-1} (x - \mu_i) - (x - c_i)^\top \Sigma_i^{-1} (x - c_i) \right| + (R + R_2)^2 \tau_2 \\
&\leq \left| (x - \mu_i)^\top \Sigma_i^{-1} (x - \mu_i) - (x - c_i)^\top \Sigma_i^{-1} (x - c_i) \right| + \epsilon/4.
\end{aligned}
$$

Continuing with the (still uncontrolled) first term,

$$
\begin{aligned}
&\left| (x - \mu_i)^\top \Sigma_i^{-1} (x - \mu_i) - (x - c_i)^\top \Sigma_i^{-1} (x - c_i) \right| \\
&= \left| (x - \mu_i)^\top \Sigma_i^{-1} (x - \mu_i) - (x - \mu_i + \mu_i - c_i)^\top \Sigma_i^{-1} (x - \mu_i + \mu_i - c_i) \right| \\
&\leq 2\|x - \mu_i\|_2 \|\mu_i - c_i\|_2 \|\Sigma^{-1}\|_2 + \|\mu_i - c_i\|_2^2 \|\Sigma^{-1}\|_2 \\
&\leq 2(R + R_2)\tau_1/\sigma_1 + \tau_1^2/\sigma_1 \\
&\leq \epsilon/4.
\end{aligned}
$$

Combining these various controls with the choices of scale parameters, for some provided probability $\alpha_i p_i$ and cover element probability $a_i p_i'$, it follows for $x \in B$ that

$$\exp(-3\epsilon/4) \leq \frac{\alpha_i p_i(x)}{a_i p_i'(x)} \leq \exp(3\epsilon/4).$$

Lastly, when $\alpha_i < \alpha_0$, simply do not bother to exhibit a cover element.

To show $|\ln(p_{\alpha,\Theta}(x)) - \ln(p_{\alpha',\Theta'}(x))| \leq \epsilon$, consider the two directions separately as follows.

1. Given the various constructions above, since $\ln$ is nondecreasing,

$$
\begin{aligned}
\ln\left( \sum_i a_i p_{\theta_i'}(x) \right) &\leq \ln\left( \sum_{\alpha_i \geq \alpha_0} \alpha_i p_{\theta_i}(x) \exp(3\epsilon/4) + \sum_{\alpha_i < \alpha_0} \alpha_i p_{\theta_i}(x) \right) \\
&\leq \ln\left( \sum_i \alpha_i p_{\theta_i}(x) \right) + \frac{3\epsilon}{4}.
\end{aligned}
$$

2. On the other hand,

$$
\begin{aligned}
\ln\left( \sum_i \alpha_i p_{\theta_i}(x) \right) &= \ln\left( \sum_{\alpha_i \geq \alpha_0} \alpha_i p_{\theta_i}(x) + \sum_{\alpha_i < \alpha_0} \alpha_i p_{\theta_i}(x) \right) \\
&\leq \ln\left( \sum_{\alpha_i \geq \alpha_0} a_i p_{\theta_i'}(x) \exp(3\epsilon/4) + k\alpha_0 p_{\max} \right) \\
&= \ln\left( (1 + \epsilon/4) \sum_{\alpha_i \geq \alpha_0} a_i p_{\theta_i'}(x) \exp(3\epsilon/4) \right. \\
&\qquad \left. - \epsilon/4 \sum_{\alpha_i \geq \alpha_0} a_i p_{\theta_i'}(x) \exp(3\epsilon/4) + k\alpha_0 p_{\max} \right).
\end{aligned}
$$

But since $\sum_i a_i \geq c_1 - k(\tau_4 + \alpha_0)$,

$$-\epsilon/4 \sum_{\alpha_i \geq \alpha_0} a_i p_{\theta_i'}(x) \exp(3\epsilon/4) \leq -(\epsilon/4)(c_1 - k(\tau_4 + \alpha_0))p_{\min} + k\alpha_0 p_{\max} \leq 0.$$

As such, since $(1 + \epsilon/4) \leq \exp(\epsilon/4)$, the result follows in this case as well.

$\square$

### D.3 Proof of Theorem 5.1

*Proof of Theorem 5.1.* This proof is based on the proof of Theorem 3.2. Let the various quantities in Lemma D.5 be given; in particular, let balls $B, C$ and their radii $R_B, R_C$ be as provided there. Additionally, define $p_0 := \exp(4c)/8e$ for convenience. Near the end of the proof, the choices $p' = p/4$ and $\epsilon := m^{-1/2+1/p}$ will be made.

By Lemma D.7, let $\mathcal{N}$ be a cover of the set $C$, with all parameters having the same names as those here, except the $R$ there is the radius $R_B$ here, and $R_2$ there is radius $R_C$ here, the lower bound $c_1$ is $p_0/p_{\max}$. By the construction of the cover there, every set of partial Gaussian parameters $(\alpha, \Theta) \in C$ with $\sum_{\alpha_i} \alpha_i \geq c_1 = p_0/p_{\max}$ and cardinality at most $k$ has a cover element $Q \in \mathcal{N}$ with

$$\sup_{x \in B} |\phi_{\mathrm{g}}(x; (\alpha, \Theta)) - \phi_{\mathrm{g}}(x; Q)| \leq \epsilon; \tag{D.8}$$

note that Lemma D.7 also provides the stated estimate of the size. Next, note for $x \in B$ and every cover element $Q \in \mathcal{N}$ that Lemma D.7 provides

$$\ln((c_1 - k\alpha_0)p_{\min}) \leq p_Q(x) \leq \ln(p_{\max})$$

where $c_1 = p_0/p_{\max}$ as above and

$$\alpha_0 = \frac{\epsilon c_1 p_{\min}}{4k(p_{\max} + \epsilon p_{\min}/2)} \leq \frac{\epsilon c_1 p_{\min}}{4k p_{\max}},$$

which combined with $\epsilon \leq 2$ and $p_{\min} \leq p_{\max}$ gives

$$c_1 - k\alpha_0 \geq c_1 \left(1 - \frac{\epsilon p_{\min}}{4p_{\max}}\right) \geq \frac{c_1}{2}.$$

Thus, by Hoeffding's inequality,

$$\sup_{Q \in \mathcal{N}} \left| \int_B \phi_{\mathrm{g}}(x; Q) d\hat{\rho}(x) - \int_B \phi_{\mathrm{g}}(x; Q) d\rho(x) \right| \leq \ln\left(\frac{p_{\max}}{p_{\min}(c_1 - k\alpha_0)}\right) \sqrt{\frac{1}{2m} \ln\left(\frac{2|\mathcal{N}|}{\delta}\right)}$$

$$\leq \ln\left(\frac{2p_{\max}^2}{p_{\min}p_0}\right) \sqrt{\frac{1}{2m} \ln\left(\frac{2|\mathcal{N}|}{\delta}\right)}. \tag{D.9}$$

For the remainder of this proof, discard the corresponding failure event.

To further simplify eq. (D.9), note firstly that

$$\ln\left(\frac{1}{p_{\min}}\right) = \ln\left((2\pi\sigma_2)^{d/2} \exp((R_B + R_C)^2/(2\sigma_1))\right)$$

$$= \ln((2\pi\sigma_2)^{d/2}) + 2R_C^2/\sigma_1,$$

where

$$R_C^2 \leq 3R_B^2(1 + \sqrt{8\sigma_2/\sigma_1})^2 + 12\sigma_2 \ln(1/\epsilon) + 6\sigma_2\left(\ln\left(\frac{64e^2(2\pi\sigma_2)^d}{(2\pi)^d p_{\max}^4}\right) - 8c\right)$$

and

$$R_B^2 \leq 2R_6^2 + M_1^2/\epsilon^{2/(p-2p')}.$$

Next, to control $|\mathcal{N}|$, the scale term $\tau = \min\{\tau_1, \tau_2\}$ must first be controlled. Since $\epsilon \leq 1$ and $\sigma_1 \leq 1$ and $R_C \geq 1$,

$$\tau \geq \frac{\epsilon \sigma_1}{16(R_B + R_C)^2} \geq \frac{\epsilon \sigma_1}{64 R_C^2},$$

and thus

$$\ln\left(\frac{\epsilon}{\tau}\right) \leq \ln(64 R_C^2/\sigma_1).$$

Together with $\tau_0 = \exp(\epsilon/4)$ and $\alpha_0 \geq \epsilon c_1 p_{\min}/(8kp_{\max}) = p_0 p_{\min}/(8kp_{\max}^2)$, and letting $\mathcal{O}(\cdot)$ swallow terms depending only on numerical constants, $c$, $\sigma_1$, and $\sigma_2$, but in particular not touching terms depending on $\epsilon$, $d$, $k$ or $m$ or $\delta$,

$$\ln(|\mathcal{N}|) \leq \ln\left(\left(\left(\frac{5}{\epsilon}\left(\frac{8kp_{\max}^2}{\epsilon p_0 p_{\min}}\right)\right)\left(\frac{3R_C d}{\tau}\right)^d \left(\frac{33}{\sigma_1 \tau}\right)^{d^2}\left(\left(\frac{\sigma_1^{-1}}{\tau/2}\right)^d + \left(\frac{\ln(\sigma_2/\sigma_1)}{\tau/2}\right)^d\right)\right)^k\right)$$

$$\leq 3d^2 k \left(5\ln(1/\epsilon) + \ln(1/p_{\min}) + 3\ln(\epsilon/\tau) + \ln(R_C) + \mathcal{O}(1)\right)$$

$$\leq 3d^2 k \left(5\ln(1/\epsilon) + 2R_C^2/\sigma_1 + 3\ln(\epsilon/\tau) + 4\ln(R_C) + \mathcal{O}(1)\right)$$

$$= \mathcal{O}\left(d^2 k(\ln(1/\epsilon) + \epsilon^{-2/(p-2p')})\right).$$

Together, the full expression in eq. (D.9) may be simplified down to

$$\sup_{Q \in \mathcal{N}} \left|\int_B \phi_g(x; Q)d\hat{\rho}(x) - \int_B \phi_g(x; Q)d\rho(x)\right|$$

$$\leq \mathcal{O}\left(\mathrm{poly}(d,k)\left(\frac{1}{\epsilon}\right)^{2/(p-2p')}\sqrt{\frac{(\ln(1/\epsilon) + (1/\epsilon)^{2/(p-2p')} + \ln(1/\delta))}{m}}\right)$$

$$\leq \mathcal{O}\left(\mathrm{poly}(d,k)\left(\frac{\epsilon^{-3/(p-2p')}}{\sqrt{m}} + \sqrt{\frac{(\ln(1/\epsilon) + \ln(1/\delta))}{m}}\right)\right)$$

$$\leq \mathcal{O}\left(\mathrm{poly}(d,k)\left(m^{-1/2+3/p} + \sqrt{\frac{(\ln(m) + \ln(1/\delta))}{m}}\right)\right) \qquad (D.10)$$

where the final step used the choice $p' = p/4$ and $\epsilon := m^{-1/2+1/p}$.

Now let any $(\alpha, \Theta) \in \mathcal{S}_{\mathrm{mog}}(\rho; c, k, \sigma_1, \sigma_2) \cup \mathcal{S}_{\mathrm{mog}}(\hat{\rho}; c, k, \sigma_1, \sigma_2)$ be given, and let $Q \in \mathcal{N}$ be a cover element satisfying eq. (D.8) for $(\alpha, \theta) \sqcap C$. By eq. (D.8), eq. (D.9), and Lemma D.5 (and thus discarding an additional failure event having probability $4\delta$),

$$\left|\int \phi_g(x; (\alpha, \Theta))d\rho(x) - \int \phi_g(x; (\alpha, \Theta))d\hat{\rho}(x)\right| \leq \left|\int \phi_g(x; (\alpha, \Theta))d\rho(x) - \int_B \phi_g(x; (\alpha, \Theta) \sqcap C)d\rho(x)\right|$$

$$+ \left|\int_B \phi_g(x; (\alpha, \Theta) \sqcap C)d\rho(x) - \int_B \phi_g(x; Q)d\rho(x)\right|$$

$$+ \left|\int_B \phi_g(x; Q)d\rho(x) - \int_B \phi_g(x; Q)d\hat{\rho}(x)\right|$$

$$+ \left|\int_B \phi_g(x; Q)d\hat{\rho}(x) - \int_B \phi_g(x; (\alpha, \Theta) \sqcap C)d\hat{\rho}(x)\right|$$

$$+ \left|\int_B \phi_g(x; (\alpha, \Theta) \sqcap C)d\hat{\rho}(x) - \int \phi_g(x; (\alpha, \Theta))d\hat{\rho}(x)\right|$$

$$\leq 4\epsilon + \ln\left(\frac{2p_{\max}^2}{p_{\min}p_0}\right)\sqrt{\frac{1}{2m}\ln\left(\frac{2|\mathcal{N}|}{\delta}\right)} + \epsilon_\rho + \epsilon_{\hat{\rho}},$$

$$= \mathrm{poly(d,k)}\mathcal{O}(m^{-1/2+3/p}\left(1 + \sqrt{\ln(m) + \ln(1/\delta)} + (1/\delta)^{4/p}\right),$$

where the final step uses the above simplification of the cover term, the choices $\epsilon = m^{-1/2+1/p}$ and $p' = p/4$, and additionally unwrapping the forms of $\epsilon_\rho$ and $\epsilon_{\hat{\rho}}$ from Lemma D.5. $\qquad \square$