[Reviews · NeurIPS 2013]

Submitted by Assigned_Reviewer_1

The paper adresses the problem of derivation of generalization bounds for the k-means objective. If the distribution is bounded, standard techniques provide good generalization bounds. Also, good bounds can be obtained under strong tail assumptions (e.g. for sub-gaussian distributions). The main focus of the current paper is to derive generalization bounds based only on moment assumptions.

The main result of the paper is that if the distribution has p\ge 4 moments, then the convergence rate is upper bounded by m^{min(-1/4,-1/2+2/p)} (here, m is the number of examples).

Evaluation:
-----------
In the realm of machine learning, boundness is fairly reasonable assumption. Nevertheless, the centrality of the k-means objective justifies research such as the one reported in this paper.

Unfortunately, I did not have enough time to read and understand the proof, which seems quite involved. Therefore, I avoid evaluating technical quality and novelty (however, as written, the results and the proofs seems non trivial). Of course, I can't vouch for correctness.

Last remark - due to the lengthy proof and the centrality of the result, I would fill more confident if the paper will undergo a journal level refereeing. This is a pity and of course unfair for the authors. Nevertheless, in the scientific community, correctness comes before fairness.
Summary: The main result seems like a truly basic result in the context of k-means generalization.

Submitted by Assigned_Reviewer_4

This paper presents finite sample excess risk bounds for k-means (and
mixtures) without the boundedness assumptions that are usually used in
this type of analysis. Although Pollard did not make boundedness
assumptions in his analysis, his results are not finite sample. The
current paper essentially turns Pollard's results into a set of finite
sample results.

The results are interesting and, although they are quite technical,
the authors make an effort to give plenty of intuition behind the
results. As far as I know, the results are new and are a welcome
addition to the known results in this area.

Minor comments:

-page 3 line 59. Should the ``1/4'' be ``-1/4''?

-I suggest adding another sentence at the beginning of Section 4
explaining the purpose of the section.
Summary: Interesting, albeit quite technical, finite sample results about k-means and mixtures.

Submitted by Assigned_Reviewer_6

The paper dis cusses the convergence rates of empirical losses of clustering to the true losses for a family of clustering tasks and losses. Results of that type have been previously published but under boundedness assumptions (of either the support of the data distribution or the loss function). Here such results are derived where bounded moments assumptions replace those boundedness restrictions.

The paper is written up in highly technical style, with almost no motivational of intuition providing comments.
All the proofs are deferred to the appendices, and I have not read them.

Assuming the results are correct, my main concern about the paper is the significance ond motivation of its results.
Summary: A technical extension of previous results on the convergence of the empirical costs of center-based clustering costs to their true distribution values. The paper lacks motivation and explations for the various definitions and results.
Author Feedback

Author rebuttal: Dear Reviewers,

Thank you for the conscientious reviews. We first discuss two general points, and then close by commenting briefly on two slightly more specific points.

1. Assigned Reviewer 6 points out insufficient motivation and explanation. To improve the presentation, we will make an editing pass with this criticism in mind, which will include the following specific adjustments.

a. We will discuss the prevalence of both heavy-tailed phenomena in nature and k-means in data mining, which together imply it is routine to encounter situations where our moment assumptions hold but the boundedness and subexponential tail assumptions of prior work fail.

b. We will make an earlier reference to the stability literature, where generalization bounds for clustering have recently received extensive treatment (and application).

c. We will make Section 2 an easier read, foreshadow the specific guarantees better, and give a better description of the parameter "c", which appears in all bounds and is innocuous; for instance, it may be set to the cost incurred by clustering with a single center at the origin.

2. Assigned Reviewer 1 points out that the conference setting leaves insufficient time for checking these proofs, and urges a journal submission. On the second point, we will certainly make a journal submission. For the first point, we think it is important that the statements and proofs are intuitive from the body of the paper alone. Therefore we will adjust the text to improve this: in particular, we can lengthen the proof sketch at the end of section 3, using up space gained by cleaning spatial excesses in the statement of Theorem 3.2 and throughout Section 2.

Lastly, regarding some specific comments:

Page 2, Line 59. This should indeed be -1/4 and not 1/4, thank you.

Section 4: Assigned Reviewer 4 suggests we add an initial sentence explaining the section. We agree, and will state that this section aims to give rates with parameters depending more obviously on natural cluster structure. We will also describe this section better in its summary on Line 95 on Page 2. We note that these comments are also relevant to the first general criticism above.